

# Comparison of turbulence measurements by a CSAT3B sonic anemometer and a high-resolution bistatic Doppler lidar

Matthias Mauder[1], Michael Eggert[2], Christian Gutsmuths[2], Stefan Oertel[2], Paul Wilhelm[2], Ingo Voelksch[1], Luise Wanner[1], Jens Tambke[3], Ivan Bogoev[4]

[1] Karlsruhe Institute of Technology, Institute of Meteorology and Climate Research, Garmisch-Partenkirchen, 82467, Germany
[2] Physikalisch-Technische Bundesanstalt, Department 1.4 Gas Flow, Braunschweig, 38116, Germany
[3] Carl von Ossietzky Universität Oldenburg, ForWind – Center for Wind Energy Research, 26129 Oldenburg, Germany
[4] Campbell Scientific Inc., Logan, UT 84321, USA

*Correspondence to*: Matthias Mauder (matthias.mauder@kit.edu)

**Abstract.** Accurate measurements of turbulence statistics in the atmosphere are important for eddy-covariance measurements, wind energy research, and the validation of atmospheric numerical models. Sonic anemometers are widely used for these applications. However, these instruments are prone to probe-induced flow distortion effects, and the magnitude of the resulting errors has been debated due to the lack of an absolute reference instrument under field conditions. Here, we present the results of an intercomparison experiment between a CSAT3B sonic anemometer and a high-resolution bistatic Doppler lidar, which is inherently free of any flow-distortion. This novel remote sensing instrument has otherwise very similar spatial and temporal sampling characteristics as the sonic anemometer and hence served as a reference for this comparison. The presented measurements were carried out over flat homogeneous terrain, at a measurement height of 30 m. We provide a comparative statistical analysis of the resulting mean wind velocities, the standard deviations of the vertical wind speed and the friction velocity and investigate the reasons for the observed deviations based on the turbulence spectra and cospectra. Our results show a very good agreement of the mean wind velocity measurements and the standard deviations of the vertical wind speed, with comparabilities of 0.082 and 0.017 m s$^{-1}$, respectively. Biases for these two quantities were very low, being smaller than 0.01 m s$^{-1}$, which corresponds to about 1 % in relative terms. Slightly larger differences were observed for friction velocity. Analysis of the corresponding cospectra showed that the CSAT3B underestimates this quantity systematically by about 3 % on average as a result of too steep a drop-off in the inertial sub-range. We also found that an angle-of-attack dependent transducer-shadowing correction does not improve this agreement effectively because it leads to an artificial correlation between the three wind components and therefore severely distorts the shape of the cospectra.

## 1 Introduction

Accurate fast-response measurements of the three-dimensional wind vector are of great importance to fundamental research in micrometeorology for flux measurements using the eddy-covariance methods in ecological studies (Aubinet et al., 2012). However, in recent years, several studies found that most, if not all, sonic anemometers may be afflicted by a systematic





underestimation of turbulent fluctuations due to probe-induced flow-distortion errors (Frank et al., 2013, 2016; Wyngaard, 1988). These errors can be further classified into errors due to transducer self-shadowing, caused by cross-shadowing and influences of the support structure. This has been demonstrated in field studies by means of specially modified reference instruments with a vertical measurement path, so that the measurement path is perfectly perpendicular to the horizontal flow,

or by rotating an additional sonic anemometer by 90° around the *x*-axis for comparison. However, the validity of these comparisons is somewhat debatable since the rotated instrument has only half of the resolution of the non-rotated for the *w*-measurement due to its vertical paths being smaller by a factor of two. Moreover, an intercomparison experiment between six different commercially available sonic anemometers showed that all participating instruments agreed very well (Mauder and Zeeman, 2018). Nevertheless, it is possible that all instruments measure vertical fluxes with similar inaccuracies, since no

independent reference measurement was available. Consequently, the absolute magnitude of the potential bias remains unknown.

A particular sonic anemometer, the CSAT3 (Campbell Scientific Inc., Logan, Utah, USA) and its variant the CSAT3B, have been investigated intensively. It is one of the most widely used and highly reputed instruments, which has often served as a reference in past intercomparison studies (Foken and Oncley, 1995; Loescher et al., 2005; Mauder et al., 2007). Features such

as its small transducer diameter, 30° tilt angle with respect to the vertical axis, short sonic path length, and symmetrical boom design following the recommendations of Wyngaard (1988) increase confidence in its high-fidelity vertical wind fluctuation measurements. Based on the results of a field comparison with an orthogonal sonic anemometer as reference, Horst et al. (2015, thereafter H15) propose a wind-tunnel derived correction for the CSAT3, which typically leads to an increase of vertical wind fluctuations and hence also vertical fluxes by 3 to 5 %.

A numerical simulation of the flow around this instrument indicates that the H15 correction actually reduced the measurement error of common turbulence statistics, but a considerable uncertainty remained (Huq et al., 2017). This study found that the error is dependent on the azimuth angle, which can be explained by cross-shadowing effects. A similar wind-direction dependence of the CSAT3's flow-distortion error was also found in a field experiment in comparison to another non-orthogonal sonic anemometer (Grare et al., 2016). Moreover, a spectral analysis based on theoretically derived ratios between the different

wind components in the inertial subrange substantiates the earlier finding that the correction by H15 only partially compensates for the CSAT3's flow distortion error (Peña et al., 2019). Nevertheless, the main problem of all these past investigations has been the lack of an accurate standard reference for the measurement of turbulent flow statistics, since wind tunnel calibrations of sonic anemometers are conducted under quasi-laminar conditions at much lower Reynolds numbers than in the free atmosphere and therefore their transferability to measurements in the field is questionable (Högström and Smedman, 2004).

Our study eliminates the limitations and uncertainties of previous experiments comparing sonic anemometers in the field. Further problems of past studies are the influence of shadowing between adjacent sensors and support structures, lack of homogenous flat terrain, and the uncertainty of the coordinate rotations. As a reference instrument, we employ a high-resolution bistatic Doppler lidar, which has been developed at the Physikalisch-Technische Bundesanstalt (PTB) in Braunschweig, Germany (Oertel et al., 2019). This optical remote sensing device is naturally free of any flow-distortion errors



and determines the 3D wind vector in a volume of less than 0.0005 m³ for measurement heights up to 200 m at an output frequency of up to 10 s$^{-1}$, which is comparable to the sampling characteristics of a typical sonic anemometer. The very small sampling volume of this lidar system has the advantage that both data sets can be directly compared, without the need for extensive modelling of spatial averaging effects, which would lead to a large uncertainty of the resulting turbulence statistics

(Brugger et al., 2016). Hence, our objectives for this study are:

- to compare the measurement of turbulence statistics of a CSAT3B sonic anemometer with the PTB lidar during a side-by-side field deployment,

- to investigate reasons for the observed deviations by means of (co-)spectral analysis, and

- to evaluate the correction proposed by H15 using the PTB lidar as a reference

In this analysis, we will mainly focus on three statistics: (i) the mean wind velocity, because this quantity is of high relevance for a number of applications, especially in wind energy research, (ii) the standard deviation of the vertical velocity component, because errors in this variable directly translate into errors of fluxes between ecosystems and the atmosphere when using the eddy-covariance method, and (iii) friction velocity, because this quantity is crucial for the validation of meteorological models (Tambke et al., 2005). To better understand the reasons for the differences between both instruments, we will analyse spectra

and cospectra of the observed turbulent time series, including an analysis of spectral ratios of wind components in the inertial subrange as proposed by Peña et al. (2019).

## 2 Methods

### 2.1 Instruments

#### 2.1.1 CSAT3B sonic anemometer

The CSAT3B sonic anemometer used in this study is the successor of the well-established CSAT3. The biggest difference to the CSAT3 is an improved placement of the control electronics inside the mounting block of the sensor head, whereas the sensor geometry, the measurement principle, etc. remained the same, so that findings of previous studies conducted with the CSAT3 are transferable to this study. The sensor geometry of the CSAT3B is after Zhang et al. (1986) which is optimized for low flow distortion due to transducer wakes designed for predominantly horizontal flow. Unlike previous sonic anemometers

with orthogonal sonic paths, where the horizontal velocity components are measured from a pair of axes located in the horizontal plane and the vertical velocity is measured by a single vertical pair of transducers, the flow-distortion effects in the CSAT3B are minimized by positioning all six transducers and their supporting structures out of the horizontal plane. In this non-orthogonal arrangement, each sonic path is tilted 30° from the vertical axis and spaced 120° apart in the horizontal plane. The length of the sonic path is 0.1154 m and the diameter of the ultrasonic transducers is 0.00635 m, giving a path length to

diameter ratio of 18. The higher this ratio the less self-shadowing effects are expected on the wind measurement, because a smaller portion of the path is affected by the transducer wake (Kaimal, 1979; Wyngaard and Zhang, 1985).





As part of the calibration procedure, the sonic path length (the distance between the transducers) and the actual values of the angles of the sonic axes of each individual CSAT3B instrument are precisely determined with a coordinate measuring machine and stored in the internal non-volatile memory. The wind speed along each sonic path is calculated from the sonic path distance between each pair of transducers and the difference of the reciprocal of the times of flight (TOF) of the ultrasonic pulses

traveling along the sonic axes in opposite directions. Accurate and precise TOF measurements are achieved using advanced digital processing techniques. The wind components along the three non-orthogonal sonic axes are transformed into orthogonal components using a 3 x 3 coordinate transformation matrix unique for each CSAT3B and derived from the actual angles determined during the geometry measurement procedure. To determine accurate TOF estimates and to account for ultrasonic transducer delays associated with the conversion of the electrical-to-acoustical signal, each CSAT3B is factory calibrated in a

specially designed temperature-controlled, zero-wind chamber over the entire operating temperature range of $-30\,°C$ to $+50\,°C$. Any temperature-induced changes in the sonic path length are also compensated for during this procedure.

The speed of sound can also be measured by the CSAT3B using the measured transducer-to-transducer distance and sum of the reciprocal of the TOF of the pulses along the acoustic path traveling in opposite direction. The quality and accuracy of the CSAT3B acoustic temperature measurements are evaluated during calibration by comparison with an air temperature standard.

This procedure provides additional independent verification of the fidelity of the TOF measurements and the accuracy of the sonic path distance.

### 2.1.2 Bistatic Doppler lidar

The most widely used wind remote sensing devices are conventional monostatic Doppler lidar systems that were established in wind energy applications in the recent years (e.g. Pearson et al., 2009). Such systems utilize a common transmitting and

receiving beam that measures the wind velocity component in beam direction via a Doppler shift of the received scattering light from aerosols traveling along the path of the transmitting laser beam (Drain, 1980). To measure the complete wind vector, the common beam is tilted in different directions (Eder et al., 2015; Newman et al., 2015). Provided that the wind field is almost homogeneous within the measurement volume, these systems deliver reliable measurement results (Gottschall et al., 2012; Peña et al., 2009). However, leaving flat terrain and having to consider the inhomogeneous wind conditions that

predominate over complex terrain, significant errors for the wind speed measured arise (Bradley, 2008) and can be on the order of 10 % (Bingöl et al., 2009). Thus, in the case of unidentified and complex wind fields, the reliability of monostatic lidar measurements becomes questionable without considering any other reference measurements.

The novel three-component lidar system developed by the PTB is aimed to overcome the present limitation to almost homogeneous wind fields given by the monostatic working principle (Oertel et al., 2019). The basic idea of this system relies

on utilizing a bistatic measurement setup (Harris et al., 2001), i.e. on the use of one transmitting laser beam and three detection beams (spatial separation), in order to determine the three components of the wind vector simultaneously in a small measurement volume by means of single aerosols (Figure 1). In contrast to monostatic systems, which typically use a common





transmitting and receiving unit and an optical circulator to separate the received scattering light, the bistatic system is based on one transmitter and three discrete, spatially separated receivers.

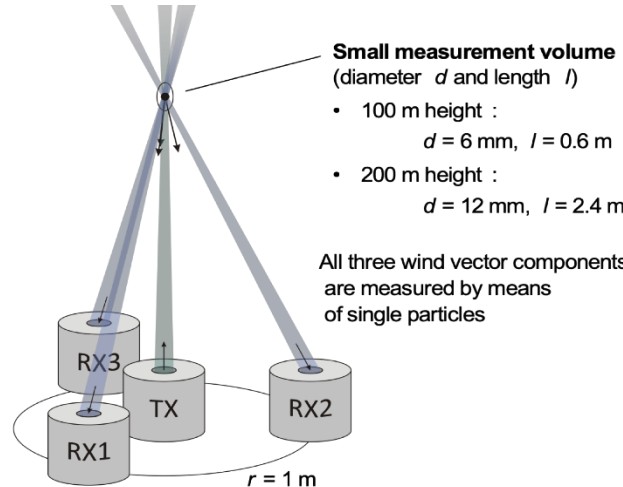

**Figure 1: Principle of the novel bistatic lidar consisting of one transmitting unit TX (green beam) and three receiving units RX**

The receivers are positioned at a radius of 1 m around the transmitter to ensure both sufficient particle-scattering light intensity (quasi-backward direction) and sufficient resolution for the determination of the horizontal velocity component. Each of the three heterodyne receivers converts the particle scattering light of its respective receiving beam into an optical beat signal, which is then converted into an electrical signal by a differential photodetector. The measurement volume calculated according

to Gaussian beam optics has a diameter of 2 mm and a length of 50 mm for a measurement height of 30 m above ground. To ensure a mobile operation with stable working conditions in the field, especially with respect to requirements on the mechanical setup and the optoelectronics, the bistatic lidar system has been enclosed in a temperature-controlled housing unit mounted on a trailer (Figure 2). The accuracy of the bistatic PTB lidar was validated with the laser Doppler anemometer (LDA) reference standard in a wind tunnel erected on a platform at a height of 8 m. Long-term measurements each lasting 1 h were carried out.

At seven velocities between 4 m s$^{-1}$ and 16 m s$^{-1}$ and different orientations of the lidar system, an average deviation of less than 0.4 % was observed.



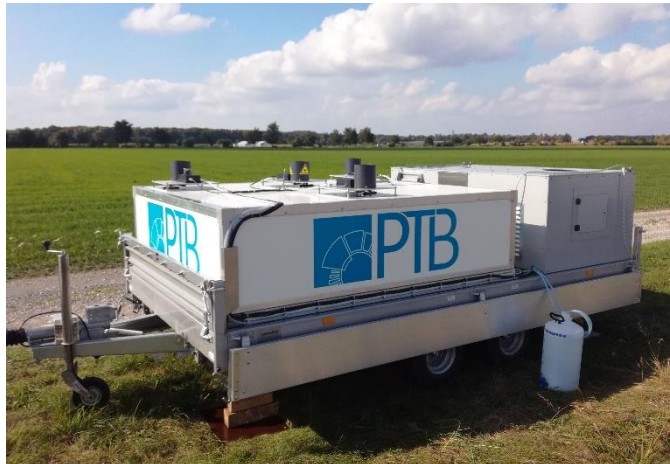

**Figure 2: Photograph of the bistatic PTB lidar at the measurement site (opened trailer housing).**

### 2.2 Experimental set-up

The field intercomparison experiment was set up at the boundary of a recently harvested maize field on the compound of the Johann Heinrich von Thünen-Institut in Braunschweig, Germany (52.2943°N 10.4461°E, 81 m a.s.l.) and measurements were carried out from 14-09-2018 0900 until 27-09-2018 0600 UTC. The CSAT3B was installed on top of a trailer-mounted pneumatic telescopic mast (Clark Masts Systems Ltd., Binstead, England) at a height of 30.5 m (Figure 3). Since the prevailing wind direction expected for the measurement period was west, the PTB lidar was set up approximately 9 m west of the trailer mast and the CSAT3B was oriented towards 270°. This set up was chosen to minimize interference from the trailer mast with the PTB lidar measurements, on the one hand, and interference from the anemometer's arms and the mast with the CSAT3B measurements, on the other. Data acquisition from the CSAT3B was accomplished using a CR6 datalogger (Campbell Scientific, Inc., Logan, Utah, USA) with SDM (Synchronous Device for Measurements) communications. The sampling rate was 10 s$^{-1}$ and the three orthogonal (referenced to the anemometer head) wind components $u_x$, $u_y$, and $u_z$ [m s$^{-1}$], the ultrasonic air temperature $T_s$ [°C] as well as the CSAT3B diagnostic flag were recorded. Measurement times were logged in UTC. The PTB lidar system recorded the measured (Doppler) frequency and amplitude of every detected scattered light signal. This raw data was also averaged to 10 s$^{-1}$ velocity vectors afterwards.





**Figure 3: Photograph of the set-up of the field intercomparison experiment between the bistatic PTB lidar (left) and the CSAT3B mounted on a mobile 30 m telescopic mast (centre right). The camera is facing north-west.**

## 2.3 Meteorological conditions

In continuation of the previous months, the air temperature stayed relatively high for the first week of the measurement campaign, due to a series of high-pressure systems. The remains of an Atlantic hurricane ("Ex-Helene") pushed hot air up to the Northern border of Germany, which culminated in air temperatures of more than 30 °C on 18 September 2018 at the site




of our experiment. The clear sky led to a strong diurnal variation in temperature with differences of up to 15 °C between the nocturnal minimum and the daytime maximum. The wind was relatively weak, 10-min mean wind speeds (in 10 m height) ranging from 1 m s$^{-1}$ and 6 m s$^{-1}$ for, and between 1 m s$^{-1}$ and 10 m s$^{-1}$ for the wind gusts. Wind speed was correlated with the variation in air temperature, with higher speeds at noon, due to more intense convection and better mixing, and lower

speeds during the night. The wind direction was mostly between South and West. At noon on 21 September 2018, the air temperature dropped abruptly from more than 25 °C to less than 15 °C, accompanied by wind speeds of up to 11 m s$^{-1}$, wind gusts of up to 20 m s$^{-1}$, and some rain. During this second week, the nocturnal temperature minimum was at 5 °C and wind speeds were generally higher than during the first week (Figure 4).

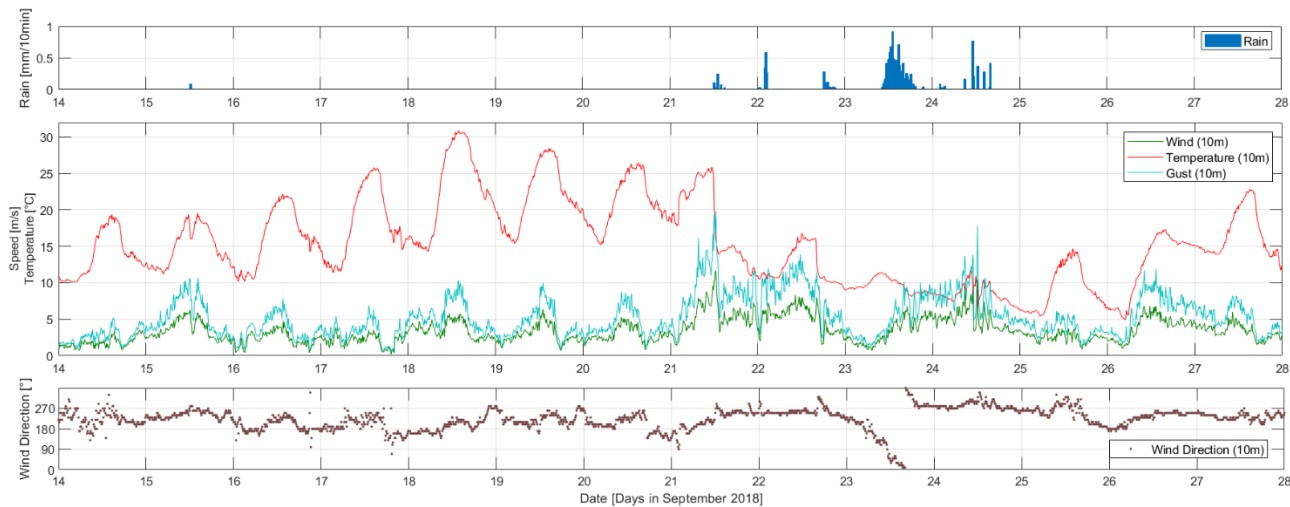

**Figure 4: Meteorological conditions during the intercomparison experiment based on 10-min data from the nearby weather station (ID 662, 52.2915° N, 10.4464° E, 81 m a.s.l.) of the German Weather Service (DWD). Data were provided by the DWD Climate Data Center (CDC).**

### 2.4 Calculation of turbulence statistics

All turbulence statistics were calculated from the 10 s$^{-1}$ raw data of both instruments using the eddy-covariance software TK3

(Mauder and Foken, 2015) with an averaging time of 30 min. The same settings were applied in TK3 for both data sets, including a spike detection algorithm (Mauder et al., 2013). In addition, we used the diagnostic flag of the CSAT3B for filtering of the raw data, and we screened our data for rain in the last hour, which may have affected the optics of the lidar and the transducers of the sonic anemometer. After this preparation of the raw data, we discarded any 3-min statistics if more than 10 % of the high-frequency data were missing. Subsequently, the raw turbulence statistics were corrected using the double

rotation (Kaimal and Finnigan, 1994), and a correction of low-pass filtering effects due to path length averaging (Moore, 1986) to allow for a direct comparison of both data sets. In an alternative processing stream, we applied the correction for transducer shadowing effects by H15 in order to validate this method as part of this intercomparison experiment. To facilitate this, we implemented this method into the TK3 software based on a software script provided by Campbell Scientific Inc.





### 2.5 Statistical analysis of the comparison

For the statistical analysis of the intercomparison, an orthogonal Deming regression was applied in order to account for measurement errors in both $x$- and $y$-variables, using the R package mcr (Manuilova et al., 2014). In this regression analysis, we generally selected the PTB lidar data as the $x$-variable and the sonic anemometer data as the y-variable. In contrast to a

traditional least-squares method, the orthogonal regression provides deviations measured perpendicularly and not parallel to the $y$-axis, which addresses problems when there is a measurement error in both $x$ and $y$ variables and implies that errors in $x$ and $y$ have equal variances. Pearson's correlation coefficient $r$ is also determined by using the same R package. Furthermore, we calculated comparability, which is equivalent to the root-mean-square-error (RMSE), and bias, which is the mean error of a certain measurement quantity.

### 2.6 Spectral analysis

Based on dimensional analysis of energy distribution of turbulence, it has been deduced that spectra and cospectra of fully developed turbulence follow similarity laws (Kolmogorov, 1941). Comparing the theoretically derived and measured spectra can be a powerful tool to investigate the performance of measuring instruments. Here, we focus on two spectral characteristics in the inertial subrange: (i) the ratio between the spectra of transversal wind velocity components, i.e. $S_v$ and $S_w$, and of the

longitudinal component $S_u$ is theoretically derived to be 4/3, and (ii) the power-law behaviour with a slope of $-5/3$ for spectra and $-7/3$ for cospectra (Kaimal and Finnigan, 1994). A ratio smaller than 4/3 between $S_w$ and $S_u$ indicates a general underestimation of vertical wind velocity or overestimation of the horizontal velocity (Peña et al., 2019). In case of high-frequency dampening, the slope of a measured spectrum drops below $-5/3$ at the high-frequency end of the spectrum (Aubinet et al., 2000). This allows for the determination of the cut-off frequency $f_c$, describing the associated sonic path averaging low-

pass filter effect, by spectral analysis as proposed by e.g. Ibrom et al. (2007). The half-hourly wind spectra are calculated using the TK3 software (Mauder and Foken, 2015), following the method of Stull (1988). Further processing is based on the method of Ibrom et al. (2007) for cut-off frequency determination. To investigate the ratio between $S_u$, $S_v$ and $S_w$ within the inertial subrange, the half-hourly spectra were weighted by frequency and exponentially binned. All $u$, $v$ and $w$ spectra from 30-minute intervals with absolute values of sensible heat flux larger than 10 W m$^{-2}$ and absolute values of the stability parameter $z/L < 2$

($z$ = measurement height, $L$ = Obukhov length) were averaged to derive one ensemble spectrum. For the empirical determination of the cut-off frequency of the $w$-measurements, half-hourly spectra $S_{w,norm}$ were additionally normalized by the variance of $w$ and inspected for blue noise.

We assume that the low-pass filtering of $S_{w,norm}(f)$ can be described by the following function (Fratini et al., 2012):

$$\frac{\frac{f \cdot S_{w,norm}(f)}{\overline{w'w'}}}{S_{w,mod}(f)} = F_n \frac{1}{1+\left(\frac{f}{f_c}\right)^2}, \tag{1}$$

where $F_n$ is an additional normalization factor, which is intended to compensate for the reduction of the overall variance (Ibrom et al., 2007). We fitted the $S_{w,norm}$ to equation (1) to determine the cut-off frequency, using the Levenberg-Marquardt nonlinear



least-squares algorithm as implemented in the R package minpack.lm (Elzhov et al., 2016). This fit was weighted by the number of frequencies in each bin. Instead of sonic temperature spectra, as proposed by Ibrom et al. (2007), we used the spectral models for vertical wind velocity $S_{w,mod}(f)$ that are implemented in TK3 as universal reference spectra. These models are a corrected version of Moore (1986) for stable stratification, and of (Højstrup, 1981) for unstable conditions. The model

spectra were calculated for each 30-minute interval and then averaged to one ensemble spectrum.

## 3 Results and discussion

### 3.1 Comparison of turbulence statistics

Scatter plots and regression parameters for $\bar{u}$, $\overline{w'w'}^{1/2}$ and $u_*$ generally show a good agreement between the CSAT3B and the PTB lidar measurements (Figure 5, Table 1). Particularly, the measurements of the vertical velocity fluctuations are almost

identical with a regression slope of 0.989, a correlation coefficient of 0.998 and a comparability of 0.017 m s$^{-1}$. This is somewhat unexpected because previous studies indicated an underestimation of $\overline{w'w'}^{1/2}$ by 3-5 % due to probe-induced flow distortion (H15, Frank et al., 2016). However, only a very small negative bias of $-0.009$ m s$^{-1}$ was found in our analysis using the flow-distortion-free PTB lidar as reference. One might argue that perhaps both instruments underestimated $\overline{w'w'}^{1/2}$ in the same way. However, we regard this as implausible, because the measurement principles are very different and therefore it is

unlikely that the effect of potential errors is so similar under this broad range of atmospheric conditions. Based on the remote optical measurement principle and the lack of any physical structure, it can safely be assumed that flow distortion errors can be ruled out for the lidar. Any potential high-frequency dampening effects of the lidar signal should be small considering a short measurement path of 0.05 m and a sampling frequency of 10 s$^{-1}$. Furthermore, we have even compensated for those small low-pass filtering effects as part of the standard post-processing routine using the TK3 software (Moore, 1986). Our findings

contradict the conclusions of earlier sonic anemometer intercomparison studies that proposed vertical wind underestimation by the CSAT3 as the source of error. The discrepancy of our findings with the results from previous experiments can be explained by the lack of a suitable and accurate reference instrument. For example, H15 used an ATI K-probe sonic anemometer as reference instrument, which they deemed to be more accurate because of its orthogonal transducer array. However, the measurements by this instrument are usually also corrected for flow-distortion effects by a factor of 1.05 on all

$w$-measurements, and this wind-tunnel based correction factor might be too large for applications in the turbulent free atmosphere.





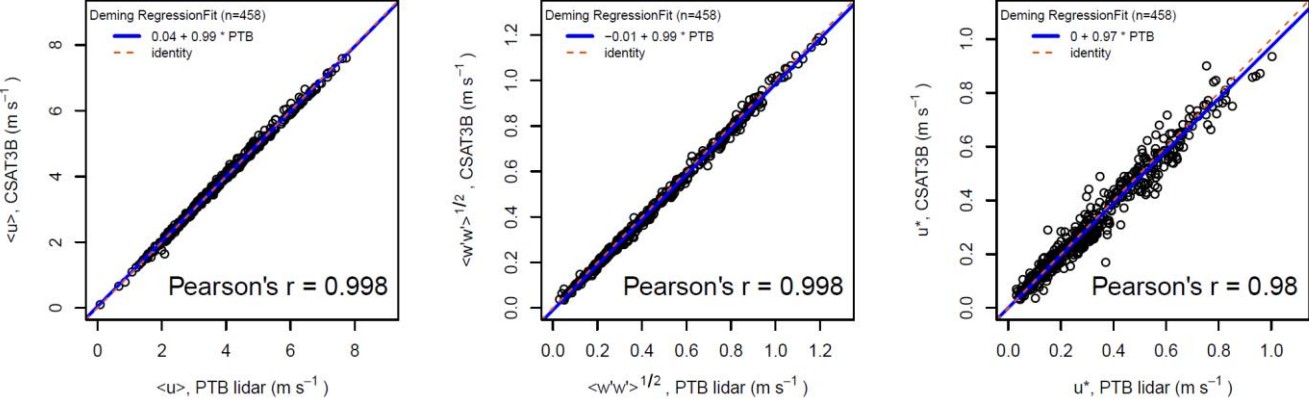

**Figure 5: Comparison for mean wind velocity (left), standard deviation of the vertical velocity component (centre), and friction velocity (right), including the regression equation and correlation coefficient. A solid blue line indicates the Deming regression and the dashed red line indicates identity.**

The mean wind velocity $\bar{u}$ also compares very well on average. There are just a few data points at higher wind speeds between 5 m s$^{-1}$ and 6 m s$^{-1}$, for which the CSAT3B reports slightly larger values than the PTB lidar. Nevertheless, this comparison, with a very small bias of 0.003 m s$^{-1}$ and a RMSE of 0.082 m s$^{-1}$ (Table 1), is still as good as or even better than between two adjacent CSAT3 sonic anemometers (Mauder and Zeeman, 2018).

Friction velocity $u_*$ is typically more difficult to measure due to the spectral separation between the peaks in the $u$ and $w$ spectra. Nevertheless, the comparability of these values is still good between the two instruments with an RMSE of 0.042 m s$^{-1}$, which is again as good as between adjacent sonic anemometers (Mauder and Zeeman, 2018). However, the $u_*$ data measured by the CSAT3B are slightly too low compared to the PTB lidar, indicated by a regression slope of 0.973 and a bias of $-0.009$ m s$^{-1}$ (Table 1). The differences in $\bar{u}$ at larger wind speeds and the systematic differences in $u_*$ will be investigated further below.

**Table 1: Statistical quantities characterizing the differences between the measurements of the CSAT3B sonic anemometer and the PTB bistatic Doppler lidar, based on 458 paired observations of turbulence statistics, each one averaged over 30 min.**

|  | $\bar{u}$ | $\overline{w'w'}^{1/2}$ | $u_*$ |
|---|---|---|---|
| **Bias** (m s$^{-1}$) | 0.003 | $-0.009$ | $-0.009$ |
| **Comparability/RMSE** (m s$^{-1}$) | 0.082 | 0.017 | 0.042 |
| **Regression intercept** (m s$^{-1}$) | 0.044 | $-0.009$ | 0.000 |
| **Regression slope** | 0.989 | 0.989 | 0.973 |
| **Correlation coefficient** | 0.998 | 0.998 | 0.980 |

As a first step, we assess whether the comparison of the CSAT3B data improves through application of the H15 method, which is intended to correct for flow distortion by transducer shadowing. As can be seen from Table 2, $\bar{u}$ and $\overline{w'w'}^{1/2}$ actually show



larger differences from the PTB lidar after applying the H15 correction. Moreover, the "corrected" mean wind velocity $\bar{u}$ has a larger bias, 0.076 instead of 0.003 m s$^{-1}$, and a larger RMSE, 0.107 instead of 0.082 m s$^{-1}$. As reported already by H15, $\overline{w'w'}^{1/2}$ is increased by 3-5 % though this correction. Our results confirm this finding, as the regression slope is increased from 0.989 to 1.030 (Table 2). However, $\overline{w'w'}^{1/2}$ is now systematically too large. Only the agreement of the $u_*$ values improves

slightly after applying the H15 correction, since the regression slope increases from 0.0973 to 1.007 and the correlation coefficient is marginally closer to unity than before (Table 2).

**Table 2: Same as Table 1, but after application of the H15 flow distortion correction to the CSAT3B data.**

|  | $\bar{u}$ | $\overline{w'w'}^{1/2}$ | $u_*$ |
|---|---|---|---|
| **Bias** (m s$^{-1}$) | 0.076 | 0.002 | −0.010 |
| **Comparability/RMSE** (m s$^{-1}$) | 0.107 | 0.016 | 0.041 |
| **Regression intercept** (m s$^{-1}$) | 0.041 | −0.011 | −0.013 |
| **Regression slope** | 1.010 | 1.030 | 1.007 |
| **Correlation coefficient** | 0.998 | 0.998 | 0.981 |

In order to investigate the reason for the remaining discrepancies in $\bar{u}$, we analysed the relationship between the differences of

the $\bar{u}$ measurements from both instruments and potential driving variables, such as $u_*$, sonic temperature, wind direction, and the standard deviations of the velocity components. We found the strongest relationship between $\Delta\bar{u}/\bar{u}$ and the wind direction (Figure 6). A very similar wind-direction dependence of the error in $\bar{u}$ has also been reported by Grare et al. (2016), when comparing a CSAT3 sonic anemometer against a Gill R3-50 sonic anemometer. It is interesting to note that this wind-direction dependence does not improve after application of the H15 flow-distortion correction (Figure 6), which only leads to larger

wind speeds in general. Hence, these results confirm the finding of Huq et al. (2017), who also found based on numerical simulations that the H15 correction does not account for the relatively pronounced azimuth dependence of the CSAT3 velocity measurements. Moreover, we can now quite reliably attribute the observed differences in $\bar{u}$ to a systematic wind-direction dependent error of the CSAT3B.





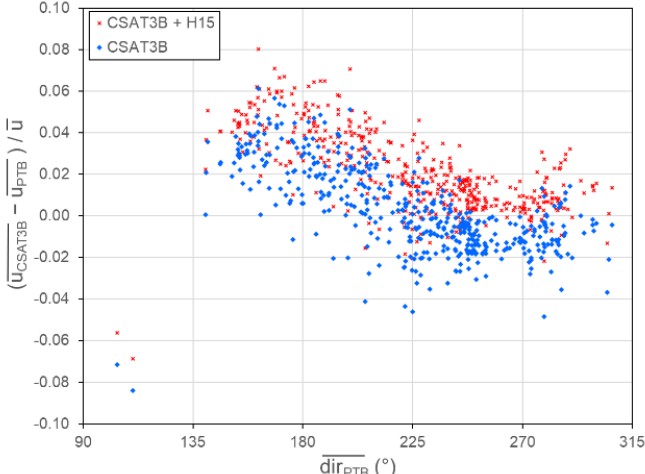

**Figure 6: Differences in wind velocity measurements between the two instruments $\overline{u}$, normalized by the mean wind velocity in $\Delta\overline{u}/\overline{u}$ versus the wind direction *dir*. The CSAT3 data are shown with and without the H15 correction.**

## 3.2 Spectral and cospectral analysis

In the following section, we investigate the ensemble turbulence spectra of the three wind components in the following section, with a special focus on the ratios between them in the inertial subrange. This may help to shed more light on the reasons for the very good agreement between the CSAT3B and the PTB lidar measurements of $\overline{w'w'}^{1/2}$. As can be seen in Figure 7, all three wind components measured by the PTB lidar are afflicted by some noise at very high frequencies. In addition, the *w*-spectra show a dampening of the signal at high frequencies. The CSAT3B spectra follow the theoretical $-5/3$ power law very well across the entire inertial subrange in all three wind components. There are no signs of noise, aliasing or high-frequency dampening in the spectra (Figure 7).





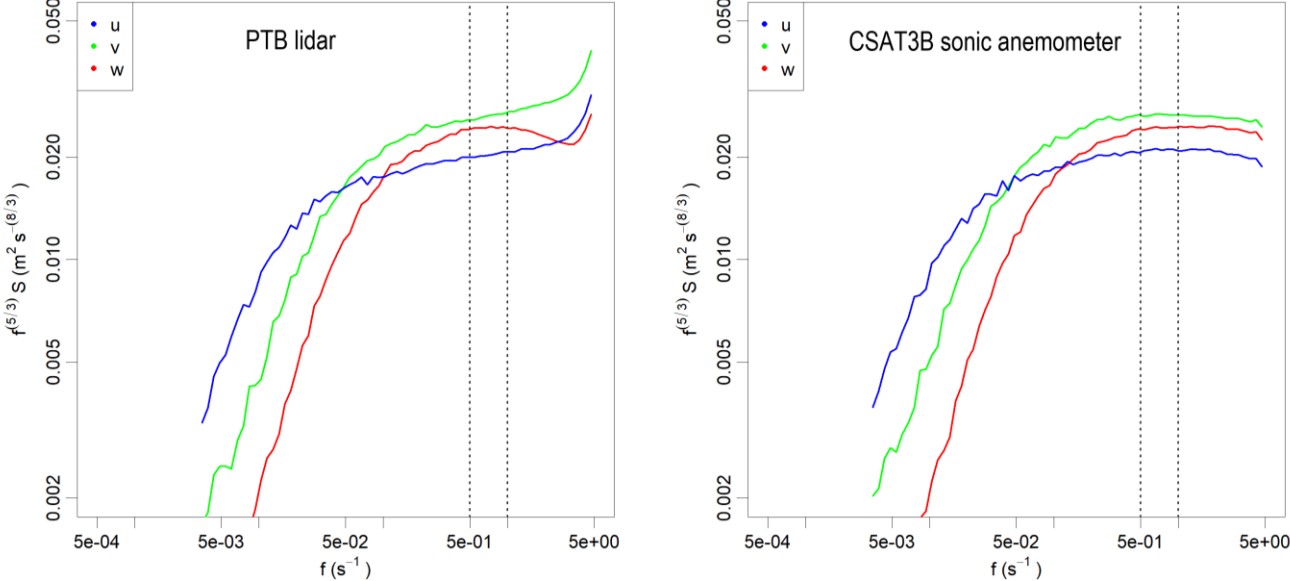

**Figure 7: Ensemble turbulence spectra of the three wind components *u*, *v*, and *w*, premultiplied by the frequency f$^{(5/3)}$, so that the theoretical −(5/3) power law appears as horizontal line. The spectra of the PTB lidar data are shown on the left and the spectra of the CSAT3B data are shown on the right. The vertical dashed lines indicate the range between 0.5 s$^{-1}$ and 1 s$^{-1}$ in the inertial subrange, for which the spectral ratios were calculated. Note that the deviations from the theoretical behaviour in the inertial subrange appear exaggerated due to the premultiplication of the spectra.**

In addition to the −5/3 power law, a spectral ratio of 4/3 has been theoretically derived for $S_v/S_u$ and $S_w/S_u$ in the inertial subrange (Kaimal and Finnigan, 1994). We generally find smaller ratios for both instruments, while the ratios measured by the PTB lidar are generally larger than for the CSAT3B by a few percent (Table 3). We also find that the $S_v/S_u$ ratios are generally larger than those for $S_w/S_u$. The lower $S_w/S_u$ ratios have been interpreted as an indicator for probe-induced flow distortion (Peña et al., 2019), which is in line with our findings since the flow-distortion free lidar measurements show larger values. However, even they do not reach the theoretical value of 4/3, neither for $S_v/S_u$ nor for $S_w/S_u$. Hence, we suspect that this theoretical value was probably not fulfilled in reality, presumably because the turbulence was not quite isotropic (Brugger et al., 2018; Stiperski and Calaf, 2017). In comparison to the uncorrected CSAT3 measurements of Peña et al. (Peña et al., 2019), our CSAT3B data show slightly smaller $S_v/S_u$ ratios of 1.26 versus 1.32 and 1.34, while the $S_w/S_u$ ratios are slightly larger, being 1.16 versus 1.13 and 1.06 for their two data sets. It is interesting to note that after the application of the H15 correction, which is supposed to correct for flow distortion effects, the spectral ratio of $S_w/S_u$ differs even more from the theoretical value of 4/3 than without the correction (Table 3). This either suggests that the 4/3 ratio is not a good criterion for detecting flow-distortion errors or, which is more likely, that the H15 method is not suitable for correcting flow-distortion effects.



**Table 3: Ratios between the spectral densities between the different wind components in the inertial subrange (at frequencies between 0.5 s⁻¹ and 1 s⁻¹) measured by the lidar and the sonic anemometer, with and without the H15 flow distortion correction. Note that both ratios $S_v/S_u$ and $S_w/S_u$ should theoretically be 4/3 assuming isotropic turbulence (Kaimal and Finnigan, 1994).**

| Spectral ratios | PTB lidar | CSAT3B | CSAT3B + H15 |
|---|---|---|---|
| $S_v/S_u$ | 1.30 | 1.26 | 1.25 |
| $S_w/S_u$ | 1.20 | 1.16 | 1.11 |

As mentioned above, all the turbulence statistics of the PTB lidar are already corrected for path-averaging effects according to Moore (1986) using a length of 0.05 m. Since the underlying analytical transfer function might not necessarily be correct, we also determined the low-pass filtering transfer function empirically based on the ensemble spectrum of $w$. We found a cut-off frequency of 4 s⁻¹, which results in an increase of $\overline{w'w'}^{1/2}$ by ca. 0.25 %, when applied as part of the Moore correction, compared to the value for the path averaging correction for 0.05 m measurement length. This small uncertainty adds confidence

to the suitability of the PTB lidar for serving as absolute reference for $\overline{w'w'}^{1/2}$ in this comparison. Significant blue noise in $S_w$ measured by the PTB lidar was not detected, either.

Obviously, the results of this intercomparison contradict the findings of H15, Frank et al. (2016) and Huq et al. (2017), who advocate the need of a flow-distortion correction on $\overline{w'w'}^{1/2}$ on the order of several percent. However, these previous field intercomparisons only compared two sonic anemometers with each other, perhaps with different sensor geometries, but none

of them can be considered as flow-distortion free as the bistatic Doppler lidar that we employed. It remains unclear why the numerical simulations of Huq et al. (2017) detect an underestimation of $\overline{w'w'}^{1/2}$ by 3 - 7 % for the CSAT3, when we see deviations of approximately 1 % in this field experiment. Perhaps the numerical simulations were not turbulent enough, so that wake effects are exaggerated, as it has also been found for wind-tunnel calibrations (Högström and Smedman, 2004). Nevertheless, our results show that an azimuth-dependent flow-distortion correction is indeed needed for obtaining more

accurate measurements of the mean wind velocity (Sect. 3.1, Figure 6).

We found that the H15 flow-distortion correction improves the $u_*$ comparison with the PTB lidar considerably, but is it accurate? An analysis of the cospectra shows that the H15 correction severely distorts the expected −7/3 power-law behaviour in the inertial subrange (Figure 8). Apparently, an artificial correlation between $u$ and $w$ is introduced at high frequencies, which can be explained by the interdependence between $u$ and $w$ introduced through this correction algorithm. Hence, our

findings support the notion of Wyngaard (1981) that the measurement of the stress tensor or friction velocity are most susceptible to probe-induced flow-distortion effects, and that any angle-of-attack based corrections are erroneous in principle. In our case, the H15 correction even results in improved $u_*$ values, but the ensemble cospectrum shows that this improvement occurred for the wrong reasons. In consequence, the observed behaviour of this correction for $u_*$ may very well be site-specific and not universally transferable. Moreover, as stated by Wyngaard (1981), such corrections are problematic because they

violate conservation of vorticity and can therefore not generally be recommended. Nevertheless, it should be noted that the





underestimation of $u_*$ measured by the CSAT3B is only by a few percent, so that the accuracy of these uncorrected measurements is still sufficient for most applications.

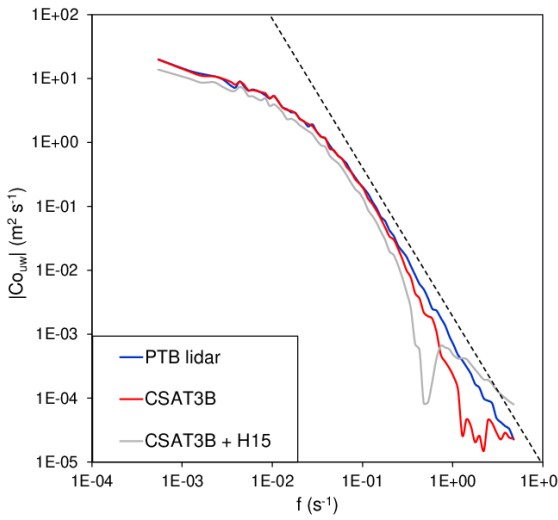

**Figure 8: Ensemble cospectra between *u* and *w* (absolute value) based on turbulence measurements from the PTB lidar and the CSAT3B sonic anemometer. The dashed line indicates the theoretical –7/3 power law in the inertial subrange.**

## 4 Conclusions

We presented the results of a field intercomparison experiment, comparing the measurements of turbulence statistics in the atmospheric surface layer of a CSAT3B sonic anemometer and a novel bistatic Doppler lidar, which has been recently developed by PTB. Spectral analysis of the high-frequency data shows that the PTB lidar has some minor noise at high frequencies in all three wind components. In addition, *w* is slightly dampened at high frequencies probably due to path length averaging, which can be corrected by a low-pass filtering correction normally applied for sonic anemometers (Moore, 1986). Nevertheless, this newly-developed instrument is well suited for serving as independent reference in measuring turbulent statistics in the atmospheric surface layer due to its traceability to laser Doppler anemometer measurements in a wind tunnel and its completely unobstructed measurement volume.

Our comparison shows a very good agreement between both instruments for the measurement of $\bar{u}$ and $\overline{w'w'}^{1/2}$. Nevertheless, our results for spectral ratios between *w* and *u* confirm that the CSAT3B is somewhat affected by flow distortion in the measurement of $\overline{w'w'}^{1/2}$. Moreover, $u_*$ from the CSAT3B is about 3 % too low compared to the PTB lidar, which is explained by too steep a drop-off of the $Co_{uw}$ cospectrum. We also evaluated whether the overall accuracy of the CSAT3B measurements can be improved by the H15 flow-distortion correction, and our results indicate that this method cannot be recommended for standard applications. It leads to a distortion of the expected power-law behaviour of $Co_{uw}$ in the inertial subrange, it leads to an overestimation of $\overline{w'w'}^{1/2}$ by approximately 3 %, and it does not correct for the wind-direction dependent error of $\bar{u}$. Hence, the probe-induced, flow-distortion issue of sonic anemometers warrants further investigations in the future.





Since any systematic effects in the measurement of $\overline{w'w'}^{1/2}$ usually directly translate into errors in eddy-covariance flux measurements, the findings of this study are also relevant with respect to the energy balance closure problem (Stoy et al., 2013) and the accuracy of any trace gas flux measurement (Foken et al., 2011; Wilson et al., 2002). In this context, we can state that the very good agreement in the $\overline{w'w'}^{1/2}$ measurements of both instruments indicates that a probe-induced flow-distortion error

of the CSAT3B sonic anemometer contributes only very little to the observed systematic underestimation of scalar fluxes using the eddy-covariance method.

In summary, the agreement of all variables tested in this comparison study is at least as good as or better than that between two adjacent sonic anemometers (Mauder and Zeeman, 2018). This indicates that both instruments are very accurate devices for measuring turbulence statistics, particularly for vertical scalar fluxes. Considering the findings of the intercomparison

experiment of Mauder and Zeeman (2018), we conclude that the other sonic anemometers tested in that study are also suitable for general flux measurements. However, our spectral analysis shows that the bistatic Doppler lidar developed by PTB is slightly more accurate, particularly for measurements of friction velocity or the momentum flux.

**Acknowledgements**

We acknowledge Mathias Herbst of the DWD Zentrum für Agrarmeteorologische Forschung in Braunschweig for providing logistical support during the measurement campaign, and we thank the Thünen-Institut in Braunschweig for providing the field site for this experiment. This study has been financially supported by the Helmholtz initiative "Modular Observations Solutions for Earth Systems (MOSES)". We thank Jamie Smidt (KIT) for checking the English grammar and spelling.

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
