# Peer review of "Comparison of turbulence measurements by a CSAT3B sonic anemometer and a high-resolution bistatic Doppler lidar"

_Atmospheric Measurement Techniques, 2019_

## Referee Comment (RC1) · Anonymous Referee #2 · 3 Jun 2019

**Summary**: This paper is focused on the question whether or not the CSAT3 sonic by Campbell Scientific needs correction for flow distortion, i.e. correction for how the instrument structure itself disturbs the flow it measures. The answer, given towards the end of the paper is a somewhat vague no, and concerning friction velocity, the authors conclude that the CSAT3 sonic anemometer is "accurate enough for most applications". The big news in the study is, however, the introduction of the bistatic Doppler lidar as a new flow-distortion free reference instrument, which can measure 3d turbulence with a very small measurement volume. It is also mentioned that the bistatic lidar can serve as a replacement for commercially available wind lidars, which typically measures the wind speed at several heights, typically in the range 50 m to 150 m. When new instruments are introduced, it is necessary that their limitation and shortcomings are discussed, but in this paper, only selected signals from the bistatic lidar are presented and no limitations with regards to accuracy and measurement capability/signal quality are mentioned. Further, the effect of the extensive postprocessing of the data is not satisfactorily described.

Whereas the study represents impressive works and achievements, I find that neither of the two main topics (#1 flow distortion in CSAT3, and #2presentation of the bi-static lidar for 3d turbulent measurements) is treated rigorously enough, and therefore recommend a major revision before being accepted for publication.

The presented conclusion that the CSAT3 sonic is different from the other sonics on the market and measures sig_w perfectly without any flow distortion correction is controversial, given earlier evidence. It is hard to overlook that other carefully designed studies have come to a different conclusion, and a more balanced discussion of the presented results in the light of the earlier studies would be an improvement.

Major criticisms:
1. When applying the flow distortion correction by Horst et al (2015) (H15), the authors find that the sig_w and U are both increased (Table 2 compared to Table 1). Yet, when presenting the cospectra in Fig 8, the covariance is decreased when applying H15 for all the low and energy containing frequencies. This is an impossible result; since the u and w components are both increased when applying H15, the absolute of the covariance must also increase for the low-frequency range. I recommend the authors to double check their algorithm and make sure that the red and grey lines have not been accidentally swapped. Since the u* comparison is improved by applying H15, the uw co-spectrum using H15 should also be closer to that by the bistatic lidar and a mistake in labelling/accidental swap seems likely.

    Further, the authors write:
    *Apparently, an artificial correlation between u and w is introduced at high frequencies, which can be explained by the interdependence between u and w introduced through this correction algorithm.*
    This argument is based on that the absolute of the cospectral density changes from around 0.00001 to around 0.0001-0.001 for high frequencies. Probably, if the absolute operator is removed, the authors would find that for the spectral range f > 1 Hz, where zero co-spectral content can be expected for the investigated setup, the sum of the co-spectral content is indeed zero and that deviations from zero are just noise. This very minor change to the spectrum is not an argument for not using H15.

2.  LL 16-18, P 14: Regarding how the H15 correction changed the spectral ratio. In H15, it is shown that the implemented transducer shadowing model has a stronger effect on *w* than on *u*. This is consistent with the difference of the results presented in Table 1 and 2. Hence, since the H15 correction is frequency independent, the correction should lead to that the spectral ratio in the inertial sublayer is increased, but here it is decreased. Again, a very strange result, please double check.

3.  When introducing a new instrument for measurement of 3D turbulence, measurements of all velocity components should be presented. Hence, the authors need to at least also show also comparisons of sig_v, sig_u and tilt angle. The latter should preferably be shown as a function of wind direction, since mismatches can result from imperfect alignment of the instruments to true vertical. Such misalignment will result in a cosine dependence of observed tilt angle to wind direction. These plots, if showing perfect agreement, can be put in an Appendix.

4.  It is truly surprising - and point to near-fantastic signal quality - that it is possible to measure the horizontal wind speed from a near vertical path. In the direction of the wind vector, the half - angle to the emitter from the receiver in the presented setup is 0.95 deg for the receiver parallel to the wind direction and less than 0.95 deg. for the two other receivers. This means that the observed Doppler shifts are very close to zero and observations near-zero have been documented to be difficult to measure accurately using Doppler lidars (Abari et al. 2015). Further, any noise mistaken for a true Doppler change should introduce a strong signal, since it results from U = V/sin(theta/2), where V is the velocity along the direction defined by the receivers and theta is the angle between the emitter and receiver. At 100 m height, as sketched in Fig. 1, the angle between the emitter and the receivers would be 0.5729 deg. When measuring a 10 m/s wind speed at 100 m height, the velocity recorded by a receiver will be lower than 5 cm/s. For 100 m, a receiver misalignment of as little as 0.01 deg. will in the system lead to a systematic error of about 1.5% in the mean wind speed estimation. What is the limitation to measurement height/measured wind speed range, in the given setup? – please explain! And how is it possible that the horizontal velocities show such little effect of noise?

5.  Data treatment: Before presentation, the data is post-processed in several steps and it is unclear what these steps do to the investigated signals. Please state/answer:
    a- how many spikes were removed in each instrument. In case there are many spikes in either of the instruments, please explain/discuss the reason for these spikes. How many removed spikes were maximum allowed in a 10 min. run?
    b- spectral treatment. The correction by Moore (1986) only corrects for path averaging on the vertical velocity component. In 2006, Horst and Oncley published exact algorithms for compensating for path-averaging for the CSAT3 geometry for all velocity components. For the spectral ratios in focus, the Horst and Oncley correction reduces the ratio (see Pena et al. Table 2) and it is therefore highly relevant for this study to implement the correct path – averaging correction. Please change/remove the Moore correction step.
    c- "After this preparation of the raw data, we discarded any 3-min statistics if more than 10 % of the high-frequency data were missing."

How many samples were removed using this step? I assume that most of the removed samples stem from the lidar?

d- It is unclear how the model spectra were used (fits to Højstrup or Moore)? Are variances and co-variances calculated from fits to model spectra?

e- Oertel et al. showed poorer agreement with reference observations for low wind speeds, but here very low wind speeds are included. What is the explanation for this improvement according to the authors?

f- The two instruments are compared in the instrument specific coordinate systems resulting from rotations V = W = 0; hence the lidar and sonic coordinate systems could be slightly different. This is from a technical perspective reasonable; it is hard to measure the exact position, yaw and tilt of each instrument. The lidar should however be easy to level, such that its w observations should indeed represent true vertical. Was leveling attempted? How different do the authors estimate the two coordinate systems to be? The latter question could be answered by plotting tilt angle versus wind direction for the two systems.

6. LL 24-27: Unlike previous sonic anemometers with orthogonal sonic paths, where the horizontal velocity components are measured from a pair of axes located in the 25 horizontal plane and the vertical velocity is measured by a single vertical pair of transducers, the flow-distortion effects in the CSAT3B are minimized by positioning all six transducers and their supporting structures out of the horizontal plane

   This is an oversimplified statement. All sonic anemometer designs suffer from flow distortion and the extent is highly dependent on the wind direction and attack angle relative to horizontal. The arrangement of transducers in the CSAT3B sonic is to my knowledge designed for a low flow distortion effect on the *horizontal* wind components and a minimized effect of white noise on the vertical velocity component. In a flux measurement, the greatest contribution of the measured flux comes from incidents of high angle of attack on the instrument (Gash and Dolman, 2003), and for high angle-of-attacks, the effects of transducer shadowing increases in the CSAT3.

7. Fig. 6: This is an interesting result, possibly indicating that the flow accelerates through the measurements volume of the CSAT3, leading to an overestimation of the horizontal wind speed. H15 hypothesized that transducer shadowing was the major cause of flow distortion in the CSAT3, which can only lead to an underestimation of the velocity, so here we are potentially looking at another major cause of flow distortion. But the presented result also leads to more questions: what kind of fluid dynamics process describes a wind acceleration without affecting all the velocity components? In other words, can an instrument that measures the vertical component perfectly (as it is claimed in this study) measure the horizontal velocity imperfectly? There could be several reasons behind the mismatch in the results; small inaccuracies in the lidar measurement height or a larger focus volume than anticipated, effects from post-processing of the data, or other inaccuracies in the optics of the lidar.

8. P. 15, LL 27-30, "In our case, the H15 correction even results in improved $u_*$ values, but the ensemble cospectrum shows that this improvement occurred for the wrong reasons. In consequence, the observed behaviour of this correction for $u_*$ may very well be site-specific and not universally transferable. Moreover, as stated by Wyngaard (1981), such corrections are problematic because they violate conservation of vorticity and can therefore not generally be recommended.". As stated above, I doubt

that the very small change in the inertial subrange will lead to a significant contribution of the u*. Moreover, the citation to Wyngaard is very strange. The mentioned tilt correction in Wyngaard (1981) has to do with the double rotation (which is used in this study) to V = W = 0. In my understanding, Wyngaard in 1981 stated that flow distortion effects cannot be avoided regardless of coordinate system. Please point to the exact place in the paper, where Wyngaard stated that flow distortion correction cannot be safely applied or remove the citation.

9. Concerning isotropy and spectral treatment. The authors should apply stricter criteria to ensure isotropy. It is not enough to select an interval where the premultiplied spectra are flat. In Pena et al (2019), a stricter selection is suggested (for example, co-spectral density should be close to zero). Moreover, it is more correct to ensemble average the spectra based on wave number instead of frequency, since the spectral content changes as a function of wind speed. The cited Stipersky and Calaf (2018) does *not* dispute that the spectra in the inertial subrange shows isotropy, but they show that this cannot be assumed for all spectra. Hence, their result is well aligned with the methods chosen in Pena et al. (2019), but not well aligned with the method for calculating spectral ratios in this paper.

Minor comments:

- L. 14, P. 6: Regarding synchronization: It is of no importance that both instruments were sampled in UTC, what matters in a 1:1 comparison is that the signals are simultaneous. Were the two instruments logged on the same data acquisition system or were both systems synchronized to GPS time? Or was synchronization attempted via the measured time series?
- Since the path-averaging is compensated for (at least for the vertical component), why do the authors expect low-pass filtering? And how should a time constant be interpreted in relation to the source of the low-pass filtering effect (path length!)?
- Boom length and mast diameter?
- Please merge Table 1 and 2, for an easier overview of the results.
- I am surprised that the CSAT spectra show no sign of noise in the high frequencies. Is it because we are looking at the ensemble *model* spectra rather than the observed spectra? Please show the original spectra.
- LL 18-19: Perhaps the numerical simulations were not turbulent enough, so that wake effects are exaggerated, as it has also been found for wind-tunnel calibrations (Högström and Smedman, 2004). It is *not* shown in Hogstrom and Smedman (2004) that wake effects are exaggerated in the wind tunnel. This is a speculation on the side of the authors. The early Gill sonics were not as good as the sonics of today. For example, three consecutive runs in a wind tunnel showed considerable scatter for the same sonic anemometer (Fig. 3 in Mortensen and Hojstrup, 1994), possibly due to changes in temperature. This could also have been a reason for why the Hogstrom and Smedman (2004) observations differed in wind tunnel and atmosphere. In any case, please correct the citation to be more precise.
- Last sentence in Abstract: We also found that an angle-of-attack dependent 25 transducer-shadowing correction does not improve this agreement effectivel because it leads to an artificial correlation between the *three* wind components and therefore severely distorts the shape of the cospectra. Only two velocity components are shown.

- LL 32, p 4: Please provide a reference for proof of the statement that the lidar measures the back scatter from each single aerosol (and not all aerosols in the measurement volume).

REFERENCES:

Abari, C. F., Pedersen, A. T., Dellwik, E., and Mann, J.: Performance evaluation of an all-fiber image-reject homodyne coherent Doppler wind lidar, Atmos. Meas. Tech., 8, 4145-4153, https://doi.org/10.5194/amt-8-4145-2015, 2015.

Gash, J.H.C. & Dolman, Han (A.J. (2003). Sonic anemometer (co)sine response and flux measurement I. The potential for (co)sine error to affect sonic anemometer-based flux measurements. Agricultural and Forest Meteorology. 119. 195–207. 10.1016/S0168-1923(03)00137-0.

Horst, T. W. and Oncley, S. P. (2006): Corrections to inertial-range power spectra measured by CSAT3 and Solent sonic anemometers, 1. Path-averaging errors, Bound.-Lay. Meteorol., 119, 375–395.

Mortensen, N. G., & Hojstrup, J. (1995). The Solent sonic - response and associated errors. In Ninth symposium on meteorological observations and instrumentation. Preprints (pp. 501-506). Boston, MA: American Meteorological Society.

---

## Referee Comment (RC2) · John Frank (Referee) · 4 Jun 2019

In this paper, a field study that compares turbulence measurements from a CSAT3 sonic anemometer and a bistatic Doppler lidar is investigated. In general, measurements from the two instruments were very similar, notably the average horizontal wind velocity and the standard deviation of the vertical wind velocity ($<w'w'>^{1/2}$, henceforth sigma_w). There was a small difference in the friction velocity u*, with the CSAT3 being lower. When a transducer shadowing algorithm was used on the CSAT3 (i.e., referred to as H15 as described by Horst et al. (2015)), all three of these measurements increased in the CSAT3, such that there was now a small difference in the vertical wind velocity with the CSAT3 being higher. An analysis of the spectral densities of the inertial subrange showed that while theoretically the Sv/Su and Sw/Su ratios should all be 4/3, only the Sv/Su ratio of the Doppler lidar was near this value. When the H15 correction was applied, the Sw/Su ratio for the CSAT3 actually decreased. The authors determine that the H15 flow-distortion correction cannot be recommended for standard applications based on the paper's results. The authors conclude that probe-induced flow distortion errors in the CSAT3 contribute little to underestimates in eddy covariance fluxes.

A big importance of this paper is that it introduces bistatic Doppler lidar measurements to sonic anemometer field studies. This is a massive step forward for micrometeorological discipline. At the same time, the results of this study appear to contradict several previous studies concerning the CSAT3. Ultimately, I do not believe this paper invalidates those studies, nor do I believe that those studies invalidate this one. Clearly, there is much more to be learned and understood about turbulence measurements, and fortunately, innovations such as the bistatic Doppler lidar help push the science forward. While I do have some major comments that should be addressed, I believe that ultimately this paper will be an extremely valuable contribution to the scientific community.

Major comments:

There are several inaccuracies in referring to Horst et al. (2015). First, on page 2, line 17-19, it states the H15 increase in vertical fluxes was 3 to 5% due to the shadowing correction algorithm. Yet, I do not find this specific range listed in that citation. Instead, that paper refers to a range of 4-5% (these values appear in their abstract and elsewhere). There is a mention in their text (i.e. the top of page 385) that sigma_w increased 3.5%. Is this where the lower value in the range 3-5% comes from? If so, this sentence should be revised to state the increase in sigma_w (i.e., 3.5%) is different from the increase in the vertical fluxes (4-5%). A concern as this sentence is written, is that it has the appearance that H15 found a smaller minimum increase in the vertical fluxes than they reported (3% versus 4%), which has the effect of diminishing the importance of H15's findings. As an extension to this, Frank et al. (2016a) calculated the increase of vertical fluxes by applying shadowing correction to a CSAT3 for a more robust set of field sites and found this ranged between 4.5-6.8% (note, these calculations were based on the original Kaimal (1979) piecewise formulation for the shadowing correction and not the Wyngaard and Zhang (1985) sinusoidal formulation that is used in H15, which in Figure 11d in Frank et al. (2016a) is demonstrated to be ~+0.6% higher). Second, the description of the reference measurement used by H15, i.e., the ATI K-probe, and its correction of 1.05 for the w measurement on page 10, lines 22-26 is incorrect. While I could not deduce the exact amount of correction applied to the K-probe data in H15, it is not possible that all w-measurements are multiplied by a fixed 1.05, or even an average value of 1.05. In Frank et al. (2016b), in which Applied Technologies (i.e. ATI) were co-authors, the specific correction for the ATI K-probe is given as a function of angle of attack. In that paper the average increase in w measurements at a Wyoming field site was ~2% (i.e., as demonstrated by the increase in the sigma_w relative difference from +1% to +3% from Tables 2 and 3). By stating that the ATI-K has a fixed w correction of 1.05, instead of a variable correction that averages ~2%, the reader is misled to believe

that the K-probe reference in H15 is fundamentally flawed, and by extension that the findings of H15 could be fundamentally flawed.

I find it troubling that in this paper the results of Huq et al. (2017) are both confirmed (i.e., page 12, line 15-17) and also condemned (page 15, lines 15-18).

I disagree with the question of validity on the Frank et al. (2016b) experiment on page 2, lines 5-7, that rotated instruments would have half the resolution which could invalidate the findings. The CSAT3 manual does specify the resolution as 0.001 m/s resolution for u and v measurements and 0.0005 m/s for w measurements (i.e., a higher resolution w-measurment). In Frank et al. (2016b) the most significant finding for the 90° rotated CSAT3 anemometers is listed in table 6, which tests that a hypothesis supporting the need for transducer shadowing would cause a -5% change in sigma_v while there would be no change in sigma_w. The observations of a -11% change in sigma_v and 0% change in sigma_w were somewhat consistent with this hypothesis. One interpretation of these results regarding measurement resolution is that the important observation that that sigma_v decreased with the 90° rotated CSAT3 anemometers was conducted with the original w-measurement path which has the higher resolution. While the authors of Frank et al. (2016b) have received criticism for their experimental design, they are unsure how issues relating to measurement resolution could invalidate their results.

The range of the results from Peña et al. (2019) appear to be misstated on page 14, line 14-16. While the values of Fv/Fu of 1.32 and 1.34 do appear in their Table 2 for the Riso and Norrekaer Enge site under CSAT3/no-correction and the value of Fw/Fu of 1.13 appears for the Riso under CSAT3/no-correction, the value they list for Norrekaer Enge site for CSAT3/no-correction is listed as 1.07 and not 1.06.

While I admittedly am new to the concept of bistatic Doppler lidar, I believe that some caution should be used before it is accepted as an unbiased control or reference measurement. First, as illustrated by the measurement volume of 2mm in horizontal diameter versus 50 mm in vertical height, this instrument clearly treats the horizontal and vertical dimensions differently. Beyond the size of the measurement volume, I assume that there is a non-orthogonal to orthogonal conversion between the measurements along the three receiving unit axes that computes the vertical measurement differently from the horizontal measurements (i.e., similar to how the CSAT3 calculates orthogonal components as described on page 4, lines 6-8). I am also troubled by Figure 7, where the spectral for the PTB lidar w measurement is clearly differently than either the u or v in the region of the inertial subrange (i.e., it is concave down while the others are ramping up). Perhaps I am not alone in questioning the use of a non-orthogonal instrument that treats the vertical dimension differently to test another non-orthogonal instrument that treats the vertical dimension differently in order to determine if there are any errors with the vertical measurement. One improvement to help address this is to present the results of the other dimensions, i.e., sigma_u, sigma_w, etc. A second improvement that could only be achieved with a new field deployment would be to collect data with the Doppler lidar focused within the CSAT3 measurement volume as well as outside of it. I once saw Tom Horst give a talk that did this with another Doppler lidar and CSAT3 study, and I recall he believed that there was a detectable difference when the lidar was focused within the path. Regardless, on page 2, line 30, it is stated that this study "eliminates the limitations" of previous studies that lacked an accurate standard. A more conservative statement is that this study seeks to improve on those limitations.

Minor comments:

Page 2, line 31-32: It is stated that there is "uncertainty of the coordinate rotations" in previous studies that is improved upon in this study. But, on page 8, line 19-21 the double coordinate rotation is implemented in this study. Does that not mean this study is also influenced by the uncertainty of coordinate rotations?

Page 5, line 13-14: It is stated that the bistatic PTB lidar is validated relative to a laser Doppler anemometer in a wind tunnel. I find this ironic since it is later stated on page 15, line 17-18 that the Huq et al. (2017) results might be exaggerated because of their relationship to the inaccuracies of wind tunnel calibrations as shown in Hogstrom and Smedman (2004).

Page 8, line 3: The word "for," might be a typo.

Page 9, Equation 1: The "," at the end of the equation might be a typo.

Page 11-12, last line/line1: The slope for u in Table 2 is actually closer to the 1:1 line than the slope for u in Table 1, so a more conservative interpretation is that the difference in u between the CSAT3 and PTB lidar does not change.

Page 12, line 1-2: While this may be the case, it is worth noting that these differences are also very small on an absolute scale.

Page 12, lines 3-6: It is interesting that the 0.041 increase in the slope of sigma_w is interpreted as "systematically too large" while the 0.034 increase of slope in u* is determined to improve "slightly". I would recommend a choice of words to emphasize that the increases in both slopes were fairly similar in size.

Page 14, Figure 7: I don't understand specifically what the last sentence in the caption is describing in the figure.

Page 14, line 14-16: I find it interesting that in a relative sense, the value of 1.26 is not that different from 1.32-1.34 while 1.16 is not that different from 1.13 and 1.06. But, in Peña et al. (2019), the difference between 1.32-1.34 and 1.07-1.13 was deemed to be evidence that there were flow distortion issues with the CSAT3 but here the difference between 1.26 and 1.16 is deemed to be evidence that there are minimal flow distortion issues with the CSAT3. It is also worth noting in Peña et al. (2019) that they present results that have the H15 correction without the path-averaging correction, but not results that have the path-averaging correction but without the H15 correction. In the case of the former, the Fw/Fu ratio actually decreases by 0.039 when the path averaging correction is applied. While I do appreciate this type of analysis, perhaps this all demonstrates that it is somewhat troublesome to interpret.

Page 15, Line 13-15: This is an incorrect statement. The main field studies of Horst et al. (2015) and Frank et al. (2016b) involved 5 simultaneously measured anemometers. If this statement is referring to the number of sonic anemometers that are simultaneously compared to each other, then the Bayesian statistical analysis in Frank et al. (2016b) simultaneously compares 13.

Page 17, line 11-12: I am not sure this is a good statement to end on, considering the spectral plot in Figure 7 shows strange behavior in the PTB lidar in the inertial subrange and the 1.20 Sw/Su ratio in Table 3 falls short of the theoretical 1.33 value.

-John Frank

References

Frank, J.M., Massman, W.J. and Ewers, B.E., 2016a. A Bayesian model to correct underestimated 3-D wind speeds from sonic anemometers increases turbulent components of the surface energy balance. Atmos. Meas. Tech., 9(12): 5933-5953.

Frank, J.M., Massman, W.J., Swiatek, E., Zimmerman, H.A. and Ewers, B.E., 2016b. All sonic anemometers need to correct for transducer and structural shadowing in their velocity measurements. Journal of Atmospheric and Oceanic Technology, 33: 149-167.

Hogstrom, U. and Smedman, A.S., 2004. Accuracy of sonic anemometers: Laminar wind-tunnel calibrations compared to atmospheric in situ calibrations against a reference instrument. Boundary-Layer Meteorology, 111(1): 33-54.

Horst, T., Semmer, S. and Maclean, G., 2015. Correction of a non-orthogonal, three-component sonic anemometer for flow distortion by transducer shadowing. Boundary-Layer Meteorology, 155(3): 371-395.

Huq, S., De Roo, F., Foken, T. and Mauder, M., 2017. Evaluation of probe-induced flow distortion of Campbell CSAT3 sonic anemometers by numerical simulation. Boundary-Layer Meteorology, 165(1): 9-28.

Kaimal, J.C., 1979. Sonic anemometer measurement of atmospheric turbulence, Proceedings of the Dynamic Flow Conference 1978 on Dynamic Measurements in Unsteady Flows. Proceedings of the Dynamic Flow Conference 1978, Skovlunde, Denmark, Skovlunde, Denmark, pp. 551-565.

Peña, A., Dellwik, E. and Mann, J., 2019. A method to assess the accuracy of sonic anemometer measurements. Atmos. Meas. Tech., 12(1): 237-252.

Wyngaard, J.C. and Zhang, S.-F., 1985. Transducer-shadow effects on turbulence spectra measured by sonic anemometers. Journal of Atmospheric and Oceanic Technology, 2(4): 548-558.

---

## Author Comment (AC1) · 26 Jul 2019

**Response to reviewer comment RC1**

The original reviewer comment are in black font and our detailed responses use blue font.

Summary: This paper is focused on the question whether or not the CSAT3 sonic by Campbell Scientific needs correction for flow distortion, i.e. correction for how the instrument structure itself disturbs the flow it measures. The answer, given towards the end of the paper is a somewhat vague no, and concerning friction velocity, the authors conclude that the CSAT3 sonic anemometer is "accurate enough for most applications". The big news in the study is, however, the introduction of the bistatic Doppler lidar as a new flow-distortion free reference instrument, which can measure 3d turbulence with a very small measurement volume. It is also mentioned that the bistatic lidar can serve as a replacement for commercially available wind lidars, which typically measures the wind speed at several heights, typically in the range 50 m to 150 m. When new instruments are introduced, it is necessary that their limitation and shortcomings are discussed, but in this paper, only selected signals from the bistatic lidar are presented and no limitations with regards to accuracy and measurement capability/signal quality are mentioned. Further, the effect of the extensive postprocessing of the data is not satisfactorily described. Whereas the study represents impressive works and achievements, I find that neither of the two main topics (#1 flow distortion in CSAT3, and #2presentation of the bi-static lidar for 3d turbulent measurements) is treated rigorously enough, and therefore recommend a major revision before being accepted for publication. The presented conclusion that the CSAT3 sonic is different from the other sonics on the market and measures sig_w perfectly without any flow distortion correction is controversial, given earlier evidence. It is hard to overlook that other carefully designed studies have come to a different conclusion, and a more balanced discussion of the presented results in the light of the earlier studies would be an improvement.

We thank the reviewer for this critical, yet very helpful, feedback and we will provide additional information where requested for further clarification. We would however respectfully disagree that this wind lidar is newly introduced in this paper at hand. As mentioned in the original manuscript version, the paper by Oertel et al. has already introduced and characterized the bistatic lidar itself. Moreover, the lidar has been presented at several instances before (see references below). In Oertel et al., the bistatic wind lidar was compared with an LDA as reference instrument in a specially designed wind tunnel. This allows the new wind lidar to be validated for wind vector measurements that are traceable to the SI units. All shortcomings and limitations of the bistatic Doppler lidar were already discussed by Oertel et al., and in the paper at hand we refer to it and repeat the main results in section 2.1.2.

Eggert, M., Müller, H. and Többen, H.: Konzeption eines Doppler-Lidar-Transfernormals zur Windgeschwindigkeitsmessung, Proceedings Lasermethoden in der Strömungsmesstechnik, 45/1 - 45/6, 2011

Eggert, M., Müller, H. and Többen, H.: Doppler-Lidar-Transfernormal zur ortsaufgelösten, vektoriellen Windgeschwindigkeitsmessung. Proceedings Lasermethoden in der Strömungsmesstechnik, 43/1 - 43/7, 2013

Eggert, M., Gutsmuths, C., Müller, H. and Többen, H.: A New Bistatic Wind LiDAR for Highly Resolved Wind Vector Measurements, Proceedings DEWEK 2015 - German Wind Energy Conference, 04.5/1 - 04.5/3, 2015

Müller, H., Eggert, M., Gutsmuths, C., Albers, A., Franke, K. and Janssen, A.-W.: A Novel Lidar System – First Results of Highly Resolved Wind Vector Measurements, Wind Europe Summit 2016. Vol. 1, 105 - 110, 2017

Major criticisms:

1. When applying the flow distortion correction by Horst et al (2015) (H15), the authors find that the sig_w and U are both increased (Table 2 compared to Table 1). Yet, when presenting the cospectra in Fig 8, the covariance is decreased when applying H15 for all the low and energy containing frequencies. This is an impossible result; since the u and w components are both increased when applying H15, the absolute of the covariance must also increase for the low-frequency range. I recommend the authors to double check their algorithm and make sure that the red and grey lines have not been accidentally swapped. Since the u* comparison is improved by applying H15, the uw co-spectrum using H15 should also be closer to that by the bistatic lidar and a mistake in labelling/accidental swap seems likely.

Indeed, sig_w, U, and u* are increased by the H15 correction. We double-checked our algorithms and the labels were displayed correctly but we actually found a bug in the averaging procedure of the CSAT3B + H15 (co-)spectra. Thanks very much to the reviewer for pointing this out! We corrected this mistake in the manuscript, and a copy of the revised Fig. 8 is presented here below for convenience:

[Figure]

Now, the grey line (CSAT3B + H15) is very close to the red line (CSAT3B) or slightly above. We would like to stress that this mistake does not affect the results of the statistical comparison.

Further, the authors write: *Apparently, an artificial correlation between u and w is introduced at high frequencies, which can be explained by the interdependence between u and w introduced through this correction algorithm*. This argument is based on that the absolute of the cospectral density changes from around 0.00001 to around 0.0001-0.001 for high frequencies. Probably, if the absolute operator is removed, the authors would find that for the spectral range f > 1 Hz, where zero cospectral content can be expected for the investigated setup, the sum of the co-spectral content is

indeed zero and that deviations from zero are just noise. This very minor change to the spectrum is not an argument for not using H15.

We agree that the red and the grey line in Fig. 8 basically just show noise for f > 1 Hz. However, the blue line (PTB lidar) does not only show noise, it actually follows the expected -7/3 power law very well up to the Nyquist frequency of 5 Hz. For clarification, we added a corresponding sentence in the revised manuscript (Sect. 3.2):

*"An analysis of the cospectra shows that the H15 correction somewhat distorts the expected −7/3 power-law behavior at very high frequencies in the inertial subrange (**Fehler! Verweisquelle konnte nicht gefunden werden.**). However, the values for f > 1 s⁻¹ are very small anyways and represent mostly white noise, which appears as horizontal line in spectral space. It can also be seen that the H15 correction slightly increases the cospectral energy across the entire range of frequencies. However, the too steep drop-off of the CSAT3B ensemble cospectrum is not improved effectively."*

We also modified the last sentence of the abstract:

*"We also found that an angle-of-attack dependent transducer-shadowing correction does not improve the already good agreement between the CSAT3B and the PTB lidar effectively."*

And we modified the corresponding sentences in the conclusions section:

*"We also evaluated whether the overall accuracy of the CSAT3B measurements can be improved by the H15 flow-distortion correction, and our results indicate that this method increases the spectral energy across the entire range of frequencies equally and does not appropriately correct the CSAT3B data in the inertial subrange. It leads to an overestimation of $\overline{(w'w')}^{(1/2)}$, and it does not correct for the wind-direction dependent error of $\overline{u}$."*

We agree that the minor change in the cospectra is not really an argument for not using H15: However, it still does not improve the cospectra effectively, it leads to an overestimation of sigma_w, which is a valid argument for not using H15 if one is interested in scalar fluxes. Therefore, we modified the conclusions as follows:

*"Based on these results, we conclude that the probe-induced flow-distortion issue of sonic anemometers warrants further investigations in the future to effectively correct general measurements of scalar fluxes."*

We decided to apply the absolute operator to the cospectra Fig 8. because the uw cospectral densities are mostly negative (since the momentum flux is negative) and they would therefore not show up at all in a logarithmic plot. We chose the logarithmic plot in order to compare the results with the -7/3 power law graphically. The non-logarithmic version of the same plot without the absolute operator is shown below for illustration.

[Figure]

2. LL 16-18, P 14: Regarding how the H15 correction changed the spectral ratio. In H15, it is shown that the implemented transducer shadowing model has a stronger effect on w than on u. This is consistent with the difference of the results presented in Table 1 and 2. Hence, since the H15 correction is frequency independent, the correction should lead to that the spectral ratio in the inertial sublayer is increased, but here it is decreased. Again, a very strange result, please double check.

Due to the abovementioned bug in the averaging algorithm for the H15 (co-)spectra, the spectral ratios were recalculated for the revised version of the manuscript. They are now in agreement with expected behavior:

*"It is interesting to note that after the application of the H15 correction, which is supposed to correct for flow distortion effects, the spectral ratio indeed agrees better with the theoretical value of 4/3 and with the PTB lidar values than without the correction (Table 3)."*

3. When introducing a new instrument for measurement of 3D turbulence, measurements of all velocity components should be presented. Hence, the authors need to at least also show also comparisons of sig_v, sig_u and tilt angle. The latter should preferably be shown as a function of wind direction, since mismatches can result from imperfect alignment of the instruments to true vertical. Such misalignment will result in a cosine dependence of observed tilt angle to wind direction. These plots, if showing perfect agreement, can be put in an Appendix.

We welcome this suggestion by the reviewer, and present the comparison plots for sigma_v and sigma_u, and the tilt angles as a function of wind direction in the Appendix of the revised version.

[Figure]

4. It is truly surprising - and point to near-fantastic signal quality - that it is possible to measure the horizontal wind speed from a near vertical path. In the direction of the wind vector, the half - angle to the emitter from the receiver in the presented setup is 0.95 deg for the receiver parallel to the wind direction and less than 0.95 deg. for the two other receivers. This means that the observed Doppler shifts are very close to zero and observations near-zero have been documented to be difficult to measure accurately using Doppler lidars (Abari et al. 2015). Further, any noise mistaken for a true Doppler change should introduce a strong signal, since it results from U = V/sin(theta/2), where V is the velocity along the direction defined by the receivers and theta is the angle between the emitter and receiver. At 100 m height, as sketched in Fig. 1, the angle between the emitter and the receivers would be 0.5729 deg. When measuring a 10 m/s wind speed at 100 m height, the velocity recorded by a receiver will be lower than 5 cm/s. For 100 m, a receiver misalignment of as little as 0.01 deg. will in the system lead to a systematic error of about 1.5% in the mean wind speed estimation. What is the limitation to measurement height/measured wind speed range, in the given setup? – please explain! And how is it possible that the horizontal velocities show such little effect of noise?

Due to the bistatic design, with no optical component reflecting light with zero Doppler frequency from the transmitter to the receivers, there is absolutely no noticeable noise increase near the carrier frequency and therefore no problem measuring low or zero Doppler shift. For sure, as in every

coherent Doppler lidar, particle signals have to be distinguished from random noise peaks. To achieve this, about 20 single spectra per receiver were sampled for each 10 Hz velocity data point and the small bandwidth of true particle signals, coming from the small measuring volume, helps to detect outliers.

It is correct that the receivers have to be precisely aligned. As they are focused to the measuring volume, an alignment error as small as 0.0034 deg in azimuth will cause the receiver to completely miss the transmitting beam while the same error in elevation will cause a measurement height offset equal to the length of the measurement volume. In order to precisely control the elevation, the optical path length is observed by some modulation of the transmitted light and a time-of-flight measurement. The air condition of the system reduces temperature induced variations of the optics alignment, allowing the measured path length to be averaged over minutes. So even at measurement heights above 100 m, with less SNR and an increased height offset to angle misalignment ratio, the uncertainty of the measurement height is in the range of a few centimeters.

We added a few words about the optics control to the revised manuscript:

*"The receivers are positioned at a radius of 1 m around the transmitter to ensure both sufficient particle-scattering light intensity (quasi-backward direction) and sufficient resolution for the determination of the horizontal velocity component. Each of the three heterodyne receivers converts the particle scattering light of its respective receiving beam into an optical beat signal, which is then converted into an electrical signal by a differential photodetector. The measurement volume calculated according to Gaussian beam optics has a diameter of 2 mm and a length of 50 mm for a measurement height of 30 m above ground. A time-of-flight measurement of the overall optical path length is used to actively control the receiver optics in order to maintain the measurement volume at the desired, well known height. To ensure a mobile operation with stable working conditions in the field, especially with respect to requirements on the mechanical setup and the optoelectronics, the bistatic lidar system has been enclosed in a temperature-controlled housing unit mounted on a trailer (**Fehler! Verweisquelle konnte nicht gefunden werden.**). …"*

5. Data treatment: Before presentation, the data is post-processed in several steps and it is unclear what these steps do to the investigated signals. Please state/answer:

a- how many spikes were removed in each instrument. In case there are many spikes in either of the instruments, please explain/discuss the reason for these spikes. How many removed spikes were maximum allowed in a 10 min. run?

We used 30 min averaging time, as it is usually done for eddy-covariance measurements, and we allowed a maximum number of 10% missing values, including those rejected by the spike test. This has been clarified in the revised version, and we also provide information about the number of spikes detected:

*"After this preparation of the raw data, we discarded any 30-min statistics if more than 10 % of the high-frequency data were missing, including those data rejected by the spike test. These are the standard settings for eddy-covariance measurements (Mauder et al. 2013, Fratini and Mauder 2014). For the CSAT3B, no spikes at all were detected for 92 % of the 618 30-min intervals, and for the PTB lidar, 73 % of the 618 30-min intervals were spike-free. This means, application of the spike detection algorithm is important to ensure high data quality, but its impact on the comparison is limited."*

b- spectral treatment. The correction by Moore (1986) only corrects for path averaging on the vertical velocity component. In 2006, Horst and Oncley published exact algorithms for compensating

for path-averaging for the CSAT3 geometry for all velocity components. For the spectral ratios in focus, the Horst and Oncley correction reduces the ratio (see Pena et al. Table 2) and it is therefore highly relevant for this study to implement the correct path – averaging correction. Please change/remove the Moore correction step.

The Moore correction was not applied on the presented spectra but only on the standard deviations and u*. Therefore, it did not affect the calculation of the spectral ratio for this study. Moreover, the implementation of the Moore correction in the TK3 software is applied to all three velocity components and the effect on the resulting standard deviations is very small as a result of the large ratio between the measurement height and the path lengths of the two instruments (>250). Pena et al. also only applied the Horst and Oncley correction to the CSAT3 data for the lower measurement height (6.4 m) and not to ones with a larger measurement height (76 m)

c- "After this preparation of the raw data, we discarded any 3-min statistics if more than 10 % of the highfrequency data were missing."How many samples were removed using this step? I assume that most of the removed samples stem from the lidar?

Please excuse the typo here, it should read 30-min statistics as in the rest of the manuscript. We report the requested information in the revised manuscript:

*"As a result of the data preparation described above, 615 30-min intervals remained for the CSAT3B and 458 for the PTB lidar."*

d- It is unclear how the model spectra were used (fits to Højstrup or Moore)? Are variances and co-variances calculated from fits to model spectra?

These model spectra were used to determine the cut-off frequency of the PTB lidar empirically. This has been clarified in the revised manuscript:

*"The model spectra were calculated for each 30-minute interval and then averaged to one ensemble spectrum in order to determine the cut-off frequency. Please note, that this does not apply to the ensemble spectra presented to determine the spectral ratios. These are purely based on measured spectra."*

e- Oertel et al. showed poorer agreement with reference observations for low wind speeds, but here very low wind speeds are included. What is the explanation for this improvement according to the authors?

In Oertel et al. we showed a direct comparison of 1 Hz velocity data measured with two instruments about three meters apart. Even at 135 m over flat terrain, this comparison obviously leads to systematic errors in the extreme values of the observed wind velocities: every time a local (temporal and spatial) minimum approaches the reference instrument, the device under test measures a higher velocity – independent of which of both instruments is the reference or the DUT and even with ideal, identical instruments. Thus, as this is an artefact of that way of a direct velocity comparison, we better analyze averages and turbulence statistics instead.

f- The two instruments are compared in the instrument specific coordinate systems resulting from rotations V = W = 0; hence the lidar and sonic coordinate systems could be slightly different. This is from a technical perspective reasonable; it is hard to measure the exact position, yaw and tilt of each instrument. The lidar should however be easy to level, such that its w observations should indeed represent true vertical. Was leveling attempted? How different do the authors estimate the two coordinate systems to be? The latter question could be answered by plotting tilt angle versus wind direction for the two systems.

The lidar system was indeed levelled, and we mostly decided to apply the double rotation method because such exact levelling is almost impossible for the sonic anemometer (Wilczak et al. 2001). Nevertheless, we applied the method also to the lidar data, because we wanted to treat both data streams in the same way. The requested plot of the tilt angle is now included in the Appendix, see also response to comment 3.

6. LL 24-27: Unlike previous sonic anemometers with orthogonal sonic paths, where the horizontal velocity components are measured from a pair of axes located in the 25 horizontal plane and the vertical velocity is measured by a single vertical pair of transducers, the flow-distortion effects in the CSAT3B are minimized by positioning all six transducers and their supporting structures out of the horizontal plane This is an oversimplified statement. All sonic anemometer designs suffer from flow distortion and the extent is highly dependent on the wind direction and attack angle relative to horizontal. The arrangement of transducers in the CSAT3B sonic is to my knowledge designed for a low flow distortion effect on the horizontal wind components and a minimized effect of white noise on the vertical velocity component. In a flux measurement, the greatest contribution of the measured flux comes from incidents of high angle of attack on the instrument (Gash and Dolman, 2003), and for high angle-of attacks, the effects of transducer shadowing increases in the CSAT3.

We agree that no sonic anemometer is free of flow distortion and we did not intent to give this impression. Nevertheless, we believe it is fair to state that the CSAT3 design reduces flow distortion to a large extend, e.g. by having a very large ratio between the path length and the transducer diameter compared to other commercially available instruments. Comparatively to other anemometers with 45 degrees tilt angels the smaller tilt angle of the CSAT3 further reduces the transducers' wake effects. To address the reviewers concern, we modified the corresponding paragraph in the revised version:

*"In comparison to previous sonic anemometers with orthogonal sonic paths, where the horizontal velocity components are measured from a pair of axes located in the horizontal plane and the vertical velocity is measured by a single vertical pair of transducers, the flow-distortion effects in the CSAT3B are reduced by positioning all six transducers and their supporting structures out of the horizontal plane. This is important because horizontal wind velocities are usually much larger than vertical wind velocities, and a distorted measurement of the horizontal wind speed directly affects the vertical wind speed measurement. In this non-orthogonal arrangement, each sonic path is tilted 30° from the vertical axis and spaced 120° apart in the horizontal plane. The length of the sonic path is 0.1154 m and the diameter of the ultrasonic transducers is 0.00635 m, giving a path length to diameter ratio of 18, which is larger than other commercially available instruments (Mauder and Zeeman, 2018). The higher this ratio and the steeper the angle between the sonic path and the vertical axis the less self-shadowing effects are expected on the wind measurement, because a smaller portion of the path is affected by the transducer wake (Kaimal, 1979; Wyngaard and Zhang, 1985)."*

7. Fig. 6: This is an interesting result, possibly indicating that the flow accelerates through the measurements volume of the CSAT3, leading to an overestimation of the horizontal wind speed. H15 hypothesized that transducer shadowing was the major cause of flow distortion in the CSAT3, which can only lead to an underestimation of the velocity, so here we are potentially looking at another major cause of flow distortion. But the presented result also leads to more questions: what kind of fluid dynamics process describes a wind acceleration without affecting all the velocity components? In other words, can an instrument that measures the vertical component perfectly (as it is claimed in this study) measure the horizontal velocity imperfectly? There could be several reasons behind the mismatch in the results; small inaccuracies in the lidar measurement height or a larger focus volume than anticipated, effects from post-processing of the data, or other inaccuracies in the optics of the lidar.

This could be explained by the horizontally symmetrical design of the CSAT3 structure as recommended by Wyngaard and Zhang (1985). In the original manuscript, we refer to the studies of Grare et al. and Huq et al. which also found a wind-direction dependent error or u_bar for the CSAT3. Moreover, Horst et al. (2016) (Boundary-Layer Meteorol DOI 10.1007/s10546-015-0123-8) observed similar behavior when they measured the flow distortion within the IRGASON integrated sonic anemometer and $CO_2$/$H_2O$ gas analyzer. They found good agreement for w, but not for U and u*. The other potential reasons for this behavior mentioned by the reviewer had already been ruled out beforehand, and they are however not discussed in the original manuscript. For example, we created the same plot based on planar-fit transformed data (instead of double rotation) and the result was quite similar. For clarification, we have modified this paragraph accordingly in the revised version:

*"This could be explained by the horizontally symmetrical design of the CSAT3 structure as recommended by Wyngaard and Zhang (1985). A very similar wind-direction dependence of the error in $\bar{u}$ has also been reported by Grare et al. (2016), when comparing a CSAT3 sonic anemometer against a Gill R3-50 sonic anemometer. Moreover, Horst et al. (2016) observed similar behavior when they measured the flow distortion within the IRGASON integrated sonic anemometer and $CO_2$/$H_2O$ gas analyzer. They found good agreement for w, but not for $\bar{u}$ and u_*."*

8. P. 15, LL 27-30, "In our case, the H15 correction even results in improved $uu*$ values, but the ensemble cospectrum shows that this improvement occurred for the wrong reasons. In consequence, the observed behaviour of this correction for $uu*$ may very well be site-specific and not universally transferable. Moreover, as stated by Wyngaard (1981), such corrections are problematic because they violate conservation of vorticity and can therefore not generally be recommended.". As stated above, I doubt that the very small change in the inertial subrange will lead to a significant contribution of the u*. Moreover, the citation to Wyngaard is very strange. The mentioned tilt correction in Wyngaard (1981) has to do with the double rotation (which is used in this study) to V = W = 0. In my understanding, Wyngaard in 1981 stated that flow distortion effects cannot be avoided regardless of coordinate system. Please point to the exact place in the paper, where Wyngaard stated that flow distortion correction cannot be safely applied or remove the citation.

As mentioned above, there was a bug in the averaging of the H15 (co-)spectra. Therefore, this paragraph is modified in the revised version, see response to comment 1. We agree that Wyngaard (1981) referred to the double rotation method when criticizing tilt corrections, and we removed this reference to Wyngaard (1981) in the revised version.

9. Concerning isotropy and spectral treatment. The authors should apply stricter criteria to ensure isotropy. It is not enough to select an interval where the premultiplied spectra are flat. In Pena et al (2019), a stricter selection is suggested (for example, co-spectral density should be close to zero). Moreover, it is more correct to ensemble average the spectra based on wave number instead of frequency, since the spectral content changes as a function of wind speed. The cited Stipersky and Calaf (2018) does not dispute that the spectra in the inertial subrange shows isotropy, but they show that this cannot be assumed for all spectra. Hence, their result is well aligned with the methods chosen in Pena et al. (2019), but not well aligned with the method for calculating spectral ratios in this paper.

We agree that it is more correct to ensemble wavenumber spectra than frequency spectra, when one is interested in characterizing turbulence. However, the main aim of this study is characterizing the instruments and therefore we used frequency spectra in this manuscript, e.g. also to determine the cut-off frequency empirically. To test the sensitivity of the results on this choice, we calculated ensemble averaged wavenumber spectra for the same range as Pena et al., Fig.3, 0.6 m-1 < k < 2 m-1.

The resulting spectral ratios are as follows. (those for the frequency spectra are in brackets for comparison):

PTB: $kSv/kSu = 1.30$ (1.30) and $kSw/kSu = 1.20$ (1.20)

CSAT3B: $kSv/kSu = 1.26$ (1.26) and $kSw/kSu = 1.14$ (1.16).

CSAT3B + H15: $kSv/kSu = 1.30$ (1.29) and $kSw/kSu = 1.23$ (1.23).

Four of these ratios are identical and two of them are slightly different. This shows that the results presented in the manuscript indeed depend on thresholds that are somewhat arbitrary. Please note that also the "sharpened criteria" of Pena et al. (2019) have somewhat arbitrary thresholds; they are just differently defined using additional criteria. Nevertheless, we find that the resulting ratio is sufficiently robust, which also shows that the inertial subrange is well-represented in the data, so that the conclusions drawn from these results are not affected, i.e. whether or not a certain ratio is close to the theoretical value 1.33 or not.

We cited Stiperski and Calaf (2018) to support our statement that spectra in the inertial subrange may not always be isotropic. If these partially anisotropic spectra are averaged, the resulting ensemble spectra must also deviate from isotropic behavior, and this is what we see in these results. For clarification, we rephrase this statement more precisely in the revised version:

*"Hence, we suspect that this theoretical value was probably not fulfilled in reality for the ensemble spectrum, presumably because the turbulence was not quite isotropic under all atmospheric conditions during the measurement period, which can happen due to different reasons (Brugger et al. 2018; Stiperski and Calaf 2018)."*

Minor comments:

• L. 14, P. 6: Regarding synchronization: It is of no importance that both instruments were sampled in UTC, what matters in a 1:1 comparison is that the signals are simultaneous. Were the two instruments logged on the same data acquisition system or were both systems synchronized to GPS time? Or was synchronization attempted via the measured time series?

The synchronization of both time series was ensured by synchronizing both data acquisition systems with a time server via internet. We clarified this in the revised version:

*"Measurement times were logged in UTC, and data acquisition systems were synchronized with a time server via internet."*

• Since the path-averaging is compensated for (at least for the vertical component), why do the authors expect low-pass filtering? And how should a time constant be interpreted in relation to the source of the low-pass filtering effect (path length!)?

Any path averaging will lead to a low-pass filtering effect. It is correct, path-averaging was compensated for. However, the corresponding correction is based on analytical transfer functions and the assumption that the path-length is correctly defined. This exercise of determining the cut-off frequency empirically was conducted to assess the sensitivity to these assumptions.

• Boom length and mast diameter?

This information has been provided in the revised version:

*"Its measuring volume was 0.85 m from the center of the mast; the mast's diameter at mounting height was 0.05 m."*

• Please merge Table 1 and 2, for an easier overview of the results.

Both tables are merged in the revised version.

• I am surprised that the CSAT spectra show no sign of noise in the high frequencies. Is it because we are looking at the ensemble model spectra rather than the observed spectra? Please show the original spectra.

These are ensemble averages of observed spectra, no model spectra are fitted here.

• LL 18-19: Perhaps the numerical simulations were not turbulent enough, so that wake effects are exaggerated, as it has also been found for wind-tunnel calibrations (Högström and Smedman, 2004). It is not shown in Hogstrom and Smedman (2004) that wake effects are exaggerated in the wind tunnel. This is a speculation on the side of the authors. The early Gill sonics were not as good as the sonics of today. For example, three consecutive runs in a wind tunnel showed considerable scatter for the same sonic anemometer (Fig. 3 in Mortensen and Hojstrup, 1994), possibly due to changes in temperature. This could also have been a reason for why the Hogstrom and Smedman (2004) observations differed in wind tunnel and atmosphere. In any case, please correct the citation to be more precise.

Here is what is stated in the abstract of Högström and Smedman (2004): "It is concluded that the correction for the effect of the vertical supporting rods of the R2 and R3 instruments, which gives nearly perfect agreement for laminar flow, does not work entirely satisfactory in the natural turbulent flow. This, in turn, is likely to be so because of high sensitivity of the wake behind the cylindrical supporting rods to the character of the approach flow."

This is in line with our statement in the original manuscript. Let us explain this reasoning further: It is well known from fluid dynamics that wake effects severely depend on the Reynolds number Re. As long as the flow is laminar, an increase in Re initially causes a growth in the size of the wake, and after transition to turbulence, a sudden reduction occurs. Wind tunnel studies are generally conducted under laminar conditions because Re is very much limited by the diameter of the wind tunnel. This means that wind-tunnel corrections are determined under conditions before this sudden reduction in wake extent at the transition from laminar to turbulent flow. Hence, they are not per se transferable to real-world turbulence, where wake effects are likely to be smaller. We explained this in the revised manuscript:

*"Perhaps the numerical simulations were not turbulent enough, so that wake effects are stronger than under fully-developed turbulent conditions in the field. Generally, wake effects depend on the Reynolds number and the wake extent is reduced suddenly at the transition from laminar to turbulent flow (e.g. Williamson 1996). This is also the reason why it is problematic to transfer quasi-laminar wind-tunnel calibrations to real-world turbulence (Högström and Smedman 2004)."*

• Last sentence in Abstract: We also found that an angle-of-attack dependent 25 transducer-shadowing correction does not improve this agreement effectivel because it leads to an artificial correlation between the three wind components and therefore severely distorts the shape of the cospectra. Only two velocity components are shown.

We removed this sentence in the revised version.

• LL 32, p 4: Please provide a reference for proof of the statement that the lidar measures the back scatter from each single aerosol (and not all aerosols in the measurement volume).

Thanks a lot for this comment, that sentence is really not precise. We actually see single particle signals at very low measurement heights, meaning with a very small measurement volume, with dimensions comparable to an LDV. In most conditions, with a larger measurement volume, the lidar receives signals from a lot of particles in the measurement volume. Anyway, the important contrast to monostatic systems is the fact that all three receiving optics receive light scattered from the same particles, so we changed that sentence:

*"The basic idea of this system relies on utilizing a bistatic measurement setup (Harris et al., 2001), i.e. on the use of one transmitting laser beam and three detection beams (spatial separation), in order to determine all three components of the wind vector simultaneously in a small measurement volume by means of the same aerosols (**Fehler! Verweisquelle konnte nicht gefunden werden.**)."*

REFERENCES:

Abari, C. F., Pedersen, A. T., Dellwik, E., and Mann, J.: Performance evaluation of an all-fiber image-reject homodyne coherent Doppler wind lidar, Atmos. Meas. Tech., 8, 4145-4153, https://doi.org/10.5194/amt-8-4145-2015, 2015.

Gash, J.H.C. & Dolman, Han (A.J. (2003). Sonic anemometer (co)sine response and flux measurement I. The potential for (co)sine error to affect sonic anemometer-based flux measurements. Agricultural and Forest Meteorology. 119. 195–207. 10.1016/S0168-1923(03)00137-0.

Horst, T. W. and Oncley, S. P. (2006): Corrections to inertial-range power spectra measured by CSAT3 and Solent sonic anemometers, 1. Path-averaging errors, Bound.-Lay. Meteorol., 119, 375–395.

Mortensen, N. G., & Hojstrup, J. (1995). The Solent sonic - response and associated errors. In Ninth symposium on meteorological observations and instrumentation.Preprints (pp. 501-506). Boston, MA: American Meteorological Society.

---

## Author Comment (AC2) · 26 Jul 2019

**Response to reviewer comment RC2**

The original reviewer comments are in black font and our detailed responses use blue font.

In this paper, a field study that compares turbulence measurements from a CSAT3 sonic anemometer and a bistatic Doppler lidar is investigated. In general, measurements from the two instruments were very similar, notably the average horizontal wind velocity and the standard deviation of the vertical wind velocity ($<w'w'>^{1/2}$, henceforth sigma_w). There was a small difference in the friction velocity u*, with the CSAT3 being lower. When a transducer shadowing algorithm was used on the CSAT3 (i.e., referred to as H15 as described by Horst et al. (2015)), all three of these measurements increased in the CSAT3, such that there was now a small difference in the vertical wind velocity with the CSAT3 being higher. An analysis of the spectral densities of the inertial subrange showed that while theoretically the Sv/Su and Sw/Su ratios should all be 4/3, only the Sv/Su ratio of the Doppler lidar was near this value. When the H15 correction was applied, the Sw/Su ratio for the CSAT3 actually decreased. The authors determine that the H15 flow-distortion correction cannot be recommended for standard applications based on the paper's results. The authors conclude that probe-induced flow distortion errors in the CSAT3 contribute little to underestimates in eddy covariance fluxes.

A big importance of this paper is that it introduces bistatic Doppler lidar measurements to sonic anemometer field studies. This is a massive step forward for micrometeorological discipline. At the same time, the results of this study appear to contradict several previous studies concerning the CSAT3. Ultimately, I do not believe this paper invalidates those studies, nor do I believe that those studies invalidate this one. Clearly, there is much more to be learned and understood about turbulence measurements, and fortunately, innovations such as the bistatic Doppler lidar help push the science forward. While I do have some major comments that should be addressed, I believe that ultimately this paper will be an extremely valuable contribution to the scientific community.

We are grateful for the insightful comment of RC2, which helped us to reconsider and reformulate some of our statements, and to provide additional information to improve clarity.

Major comments:

There are several inaccuracies in referring to Horst et al. (2015).

First, on page 2, line 17-19, it states the H15 increase in vertical fluxes was 3 to 5% due to the shadowing correction algorithm. Yet, I do not find this specific range listed in that citation. Instead, that paper refers to a range of 4-5% (these values appear in their abstract and elsewhere). There is a mention in their text (i.e. the top of page 385) that sigma_w increased 3.5%. Is this where the lower value in the range 3-5% comes from? If so, this sentence should be revised to state the increase in sigma_w (i.e., 3.5%) is different from the increase in the vertical fluxes (4-5%). A concern as this sentence is written, is that it has the appearance that H15 found a smaller minimum increase in the vertical fluxes than they reported (3% versus 4%), which has the effect of diminishing the importance of H15's findings.

Here are the two sentences from H15, which led us to summarize their results in this way with a range of 3 – 5%:

"Our simulations of transducer shadowing with the CSAT3 path geometry using the HATS dataset find that the attenuation of wt equals 5 % independent of stability, and our Marshall sonic

intercomparison data suggest averaged over all wind directions an attenuation of 3–4 % averaged over 6 months of data."

However, we agree with RC2 that it is better to quote the range given in the abstract, which is 4-5%, and we corrected this in the revised manuscript.

As an extension to this, Frank et al. (2016a) calculated the increase of vertical fluxes by applying shadowing correction to a CSAT3 for a more robust set of field sites and found this ranged between 4.5-6.8% (note, these calculations were based on the original Kaimal (1979) piecewise formulation for the shadowing correction and not the Wyngaard and Zhang (1985) sinusoidal formulation that is used in H15, which in Figure 11d in Frank et al. (2016a) is demonstrated to be ~+0.6% higher).

Thanks for this additional information.

Second, the description of the reference measurement used by H15, i.e., the ATI K-probe, and its correction of 1.05 for the w measurement on page 10, lines 22-26 is incorrect. While I could not deduce the exact amount of correction applied to the K-probe data in H15, it is not possible that all w-measurements are multiplied by a fixed 1.05, or even an average value of 1.05. In Frank et al. (2016b), in which Applied Technologies (i.e. ATI) were co-authors, the specific correction for the ATI K-probe is given as a function of angle of attack. In that paper the average increase in w measurements at a Wyoming field site was ~2% (i.e., as demonstrated by the increase in the sigma_w relative difference from +1% to +3% from Tables 2 and 3). By stating that the ATI-K has a fixed w correction of 1.05, instead of a variable correction that averages ~2%, the reader is misled to believethat the K-probe reference in H15 is fundamentally flawed, and by extension that the findings of H15 could be fundamentally flawed.

Thanks for this important information. The effect of the ATI-K probe's flow distortion correction was not clearly stated in H15 and we had relied on third party information. We changed the corresponding sentence in the revised manuscript accordingly:

*"H15 used an ATI K-probe sonic anemometer as reference instrument, which they assumed to be more accurate because of its orthogonal transducer array. However, the measurements by this instrument are also corrected for flow-distortion effects by a variable factor of 1.02 on average for w-measurements, and this wind-tunnel based correction factor might not be applicable in the turbulent free atmosphere."*

Please note that even H15 state in the last paragraph of their conclusion section that it is a shortcoming of their study to use another sonic anemometer as reference:

"The principal shortcoming of our research is the dependence of the results on a comparison between sonics … Thus the supporting evidence for our proposed correction is somewhat indirect and incomplete. We have assumed that vertical velocity measurements made with a dedicated vertical path, such as with the ATI-K sonic, are a valid reference standard for the CSAT3 measurements."

Moreover, in this study at hand, we followed the call to action of H15 at the end of their conclusion section:

"The ideal evidence for our proposal would be a comparison of sonic anemometer measurements to a reference that is free from flow distortion. One promising technique is that employed by Dellwik et al. (2015), who made simultaneous velocity measurements with a CSAT3 and with a three-component Doppler lidar system"

I find it troubling that in this paper the results of Huq et al. (2017) are both confirmed (i.e., page 12, line 15-17) and also condemned (page 15, lines 15-18).

Indeed, the results of this study indicate that the qualitative finding of an azimuth dependence of the CSAT3 error is confirmed (which has also already been found by Grare et al.), but quantitative magnitude of the underestimation for sigma_w is obviously overestimated by Huq et al. (2017). We base this assessment on the assumption that the flow-distortion free measurements of the PTB lidar in real-world turbulence are more reliable as a reference than the numerical simulations with fluctuating but not fully turbulent inflow.

I disagree with the question of validity on the Frank et al. (2016b) experiment on page 2, lines 5-7, that rotated instruments would have half the resolution which could invalidate the findings. The CSAT3 manual does specify the resolution as 0.001 m/s resolution for u and v measurements and 0.0005 m/s for w measurements (i.e., a higher resolution w-measurment). In Frank et al. (2016b) the most significant finding for the 90° rotated CSAT3 anemometers is listed in table 6, which tests that a hypothesis supporting the need for transducer shadowing would cause a -5% change in sigma_v while there would be no change in sigma_w. The observations of a -11% change in sigma_v and 0% change in sigma_w were somewhat consistent with this hypothesis. One interpretation of these results regarding measurement resolution is that the important observation that that sigma_v decreased with the 90° rotated CSAT3 anemometers was conducted with the original w-measurement path which has the higher resolution. While the authors of Frank et al. (2016b) have received criticism for their experimental design, they are unsure how issues relating to measurement resolution could invalidate their results.

We agree with RC2 that this was not a good argument, and this statement has been removed in the revised version.

The range of the results from Peña et al. (2019) appear to be misstated on page 14, line 14-16. While the values of Fv/Fu of 1.32 and 1.34 do appear in their Table 2 for the Riso and Norrekaer Enge site under CSAT3/no-correction and the value of Fw/Fu of 1.13 appears for the Riso under CSAT3/no-correction, the value they list for Norrekaer Enge site for CSAT3/no-correction is listed as 1.07 and not 1.06.

Thanks, we corrected this typo and replaced the 1.06 by 1.07.

While I admittedly am new to the concept of bistatic Doppler lidar, I believe that some caution should be used before it is accepted as an unbiased control or reference measurement. First, as illustrated by the measurement volume of 2mm in horizontal diameter versus 50 mm in vertical height, this instrument clearly treats the horizontal and vertical dimensions differently. Beyond the size of the measurement volume, I assume that there is a non-orthogonal to orthogonal conversion between the measurements along the three receiving unit axes that computes the vertical measurement differently from the horizontal measurements (i.e., similar to how the CSAT3 calculates orthogonal components as described on page 4, lines 6-8). I am also troubled by Figure 7, where the spectral for the PTB lidar w measurement is clearly differently than either the u or v in the region of the inertial subrange (i.e., it is concave down while the others are ramping up). Perhaps I am not alone in questioning the use of a nonorthogonal instrument that treats the vertical dimension differently to test another non-orthogonal instrument that treats the vertical dimension differently in order to determine if there are any errors with the vertical measurement. One improvement to help address this is to present the results of the other dimensions, i.e., sigma_u, sigma_w, etc. A second improvement that could only be achieved with a new field deployment would be to collect data with the Doppler lidar focused within the CSAT3 measurement volume as well as outside of it. I once saw Tom Horst give a talk that did this with another Doppler lidar and CSAT3 study, and I recall he

believed that there was a detectable difference when the lidar was focused within the path. Regardless, on page 2, line 30, it is stated that this study "eliminates the limitations" of previous studies that lacked an accurate standard. A more conservative statement is that this study seeks to improve on those limitations.

Thanks, we reformulated this statement in a more conservative way, saying now that this study seeks to overcome the limitations. We also considered measuring with the lidar within the measurement volume of the sonic anemometer, but we realized that measurements in the nearby undisturbed flow are what is needed to characterize an instrument, as it has also been done by Huq et al. (2017) in their numerical experiment and as it has been expressed in the last paragraph of H15.

Minor comments:

Page 2, line 31-32: It is stated that there is "uncertainty of the coordinate rotations" in previous studies that is improved upon in this study. But, on page 8, line 19-21 the double coordinate rotation is implemented in this study. Does that not mean this study is also influenced by the uncertainty of coordinate rotations?

We agree and we removed this statement. We also processed the data in natural coordinates and planar fit coordinates, and found more or less the same results. But we had to make a choice and decided to apply the double rotation method. By the way, the tilt angles are now presented as a function of wind direction in the Appendix of the revised version.

Page 5, line 13-14: It is stated that the bistatic PTB lidar is validated relative to a laser Doppler anemometer in a wind tunnel. I find this ironic since it is later stated on page 15, line 17-18 that the Huq et al. (2017) results might be exaggerated because of their relationship to the inaccuracies of wind tunnel calibrations as shown in Hogstrom and Smedman (2004).

It is absolutely no problem to validate a flow-distortion free remote-sensing instrument, such as a Doppler lidar, in a wind tunnel, since it has no wake effects that might be affected by the difference in Reynolds number between the wind tunnel (quasi-laminar) and the free atmosphere (highly turbulent).

Page 8, line 3: The word "for," might be a typo.

Thanks, the word "for" has been removed.

Page 9, Equation 1: The "," at the end of the equation might be a typo.

This comma introduces the following subclause starting with "where …". However we added an additional space between the comma and the equation to clarify that it is not part of the equation.

Page 11-12, last line/line1: The slope for u in Table 2 is actually closer to the 1:1 line than the slope for u in Table 1, so a more conservative interpretation is that the difference in u between the CSAT3 and PTB lidar does not change.

We agree that slope and intercept are similar after applying the H15 correction. Nevertheless, bias and RMSE are clearly increased. Hence, we modified this sentence in the revised version:

*"Moreover, the "corrected" mean wind velocity $\bar{u}$ has a larger bias, 0.076 instead of 0.003 m s−1, and a larger RMSE, 0.107 instead of 0.082 m s−1, although intercept and slope are similar to before applying the H15 correction."*

Page 12, line 1-2: While this may be the case, it is worth noting that these differences are also very small on an absolute scale.

We agree and we modified this sentence in the revised version by adding the word "slightly":

"as can be seen from Table 1, $\bar{u}$ and $\overline{w'w'}^{1/2}$ show _slightly_ larger differences from the PTB lidar after applying the H15 correction"

Page 12, lines 3-6: It is interesting that the 0.041 increase in the slope of sigma_w is interpreted as "systematically too large" while the 0.034 increase of slope in u* is determined to improve "slightly". I would recommend a choice of words to emphasize that the increases in both slopes were fairly similar in size.

We agree, that these formulations might lead to a misunderstanding with respect to the effect of the H15 correction and we rephrased the corresponding sentences:

"H15 reported that $(w'w')^{(1/2)}$ is increased by 4-5 % though this correction. Our results are on the lower end of this range, as the regression slope is increased from 0.989 to 1.030 (Table 2). However, the slope is now clearly larger than unity and the regression intercept for $(w'w')^{(1/2)}$ slightly more negative, so that the comparability is almost identical before and after the correction. The agreement of the $u_*$ values improves slightly after applying the H15 correction, since the regression slope increases from 0.0973 to 1.007 and the correlation coefficient is marginally closer to unity than before (Table 1)."

Page 14, Figure 7: I don't understand specifically what the last sentence in the caption is describing in the figure.

The spectra are multiplied with f^5/3, and this factor is increasing rapidly towards higher frequencies. Therefore, deviations from the expected flat behavior appear larger at high frequencies that at low frequencies in the inertial subrange. We changed this sentence slightly in the revised version for clarification:

"Note that the deviations from the expected behavior in the inertial subrange appear larger than at lower frequencies due to the premultiplication."

Page 14, line 14-16: I find it interesting that in a relative sense, the value of 1.26 is not that different from 1.32-1.34 while 1.16 is not that different from 1.13 and 1.06. But, in Peña et al. (2019), the difference between 1.32-1.34 and 1.07-1.13 was deemed to be evidence that there were flow distortion issues with the CSAT3 but here the difference between 1.26 and 1.16 is deemed to be evidence that there are minimal flow distortion issues with the CSAT3. It is also worth noting in Peña et al. (2019) that they present results that have the H15 correction without the path-averaging correction, but not results that have the path-averaging correction but without the H15 correction. In the case of the former, the Fw/Fu ratio actually decreases by 0.039 when the path averaging correction is applied. While I do appreciate this type of analysis, perhaps this all demonstrates that it is somewhat troublesome to interpret.

We would agree that the spectral ratios analysis alone makes it difficult to assess whether a sensor is affected by flow distortion or not, or whether a correction, either for path averaging or flow distortion effect, really improves the accuracy of the measurements. In our study, we have a flow-distortion free reference instrument, which measures in an even smaller volume that the sonic anemometer and at the same temporal resolution. Moreover, it can be traced back to SI standards. No measurement device is ideal, but because of these superior characteristics makes the PTB lidar a

very good reference instrument, and because of the good agreement with this reference instrument, we conclude that flow distortion effects of the CSAT3B are not as severe as expected and that the H15 correction does not effectively correct for the remaining flow distortion effects. Another problem is that even the flow-distortion free lidar data are not fully in agreement with the theoretical value of 4/4 for sigma_w. We have slightly rephrased this statement in the revised version for clarification:

Hence, we suspect that this theoretical value was probably not fulfilled in reality for the ensemble spectrum, presumably because the turbulence was not quite isotropic under all atmospheric conditions during the measurement period:

*"However, these flow-distortion free data do not reach the theoretical value of 4/3, neither for $S_v/S_u$ and even less for $S_w/S_u$. Hence, we suspect that this theoretical value was probably not fulfilled in reality for the ensemble spectrum, presumably because the turbulence was not quite isotropic under all atmospheric conditions during the measurement period."*

Page 15, Line 13-15: This is an incorrect statement. The main field studies of Horst et al. (2015) and Frank et al. (2016b) involved 5 simultaneously measured anemometers. If this statement is referring to the number of sonic anemometers that are simultaneously compared to each other, then the Bayesian statistical analysis in Frank et al. (2016b) simultaneously compares 13.

The number of sonic anemometers is not really relevant here, but we agree that it was of course more than two, and we rephrased this sentence accordingly:

*"However, these previous field intercomparisons only compared different sonic anemometers with each other, partially with different sensor geometries, but none of them can be considered as flow-distortion free as the bistatic Doppler lidar."*

Page 17, line 11-12: I am not sure this is a good statement to end on, considering the spectral plot in Figure 7 shows strange behavior in the PTB lidar in the inertial subrange and the 1.20 Sw/Su ratio in Table 3 falls short of the theoretical 1.33 value.

If the PTB lidar does not fulfill the theoretical 1.33 value, this can have theoretically three reasons.

a) Flow distortion, which can be ruled out because this is a remote sensing instrument
b) path averaging, which is expected to be small due to the very small measurement volume, and which was additionally ruled out to be significant by the empirical determination of the cut-off frequency.
c) The theory of isotropic turbulence does not fully apply to all 30-min intervals of this intercomparison experiment.

Since explanation a) and b) are ruled out, we believe that the deviations from the theoretical value of 4/3 are real. Note, that the deviations from the expected spectral behavior are enlarged in the inertial subrange in Figure 7 due to the premultiplication with f^5/3, as explained above. We modified this last paragraph slightly, just to express more precisely what we intend to say:

*"In summary, the agreement of all variables tested in this comparison experiment is at least as good as or better than that between two adjacent sonic anemometers (Mauder and Zeeman, 2018). This indicates that both instruments are very precise devices for measuring turbulence statistics, particularly for vertical scalar fluxes. Considering the findings of the intercomparison experiment of Mauder and Zeeman (2018), we conclude that the other sonic anemometers tested in that study are also suitable for general flux measurements within the range of comparability and bias described in*

*that study. However, our spectral analysis shows that the bistatic Doppler lidar developed by PTB is slightly more accurate, particularly for measurements of friction velocity or the momentum flux."*

-John Frank

References

Frank, J.M., Massman, W.J. and Ewers, B.E., 2016a. A Bayesian model to correct underestimated 3-D wind speeds from sonic anemometers increases turbulent components of the surface energy balance. Atmos. Meas. Tech., 9(12): 5933-5953.

Frank, J.M., Massman, W.J., Swiatek, E., Zimmerman, H.A. and Ewers, B.E., 2016b. All sonic anemometers need to correct for transducer and structural shadowing in their velocity measurements. Journal of Atmospheric and Oceanic Technology, 33: 149-167.

Hogstrom, U. and Smedman, A.S., 2004. Accuracy of sonic anemometers: Laminar wind-tunnel calibrations compared to atmospheric in situ calibrations against a reference instrument. Boundary-Layer Meteorology, 111(1): 33-54.

Horst, T., Semmer, S. and Maclean, G., 2015. Correction of a non-orthogonal, three-component sonic anemometer for flow distortion by transducer shadowing. Boundary-Layer Meteorology, 155(3): 371-395.

Huq, S., De Roo, F., Foken, T. and Mauder, M., 2017. Evaluation of probe-induced flow distortion of Campbell CSAT3 sonic anemometers by numerical simulation. Boundary-Layer Meteorology, 165(1): 9-28.

Kaimal, J.C., 1979. Sonic anemometer measurement of atmospheric turbulence, Proceedings of the Dynamic Flow Conference 1978 on Dynamic Measurements in Unsteady Flows. Proceedings of the Dynamic Flow Conference 1978, Skovlunde, Denmark, Skovlunde, Denmark, pp. 551-565.

Peña, A., Dellwik, E. and Mann, J., 2019. A method to assess the accuracy of sonic anemometer measurements. Atmos. Meas. Tech., 12(1): 237-252.

Wyngaard, J.C. and Zhang, S.-F., 1985. Transducer-shadow effects on turbulence spectra measured by sonic anemometers. Journal of Atmospheric and Oceanic Technology, 2(4): 548-558.

---

## Referee Report (RR1)

Remaining criticism:

The authors have responded well to most of the criticism, and errors have been corrected, which is good.  I remain truly impressed by the PTB lidar, but I still see a few potential errors in the paper.

- Concerning stating uncertainties in the PTB lidar. All papers should address the main uncertainties in the presented data; this is standard scientific good practice. Given that the authors present somewhat inconsistent results (see below), it would be interesting for the readers to learn, whether there is a possible weak point. In example, the authors write in the review answer that an error as small as 0.0034deg in the azimuth is a huge problem.  What is their pointing accuracy and how does it translate to wind speed errors?

- Abstract:
  "Analysis of the corresponding cospectra showed that the CSAT3B underestimates this quantity systematically by about 3 % on average ==as a result of too steep a drop-off in the inertial subrange==."
  The authors still claim that the difference in momentum flux between the PTB lidar and the sonic anemometer is due to the clearly seen difference in the spectra between 0.1 and 5Hz in Figure 8. However, as stated in my original review, because of the double log-axis plot, this way of illustrating error could be very misleading. In order to let the reader be able to see the frequency dependent contribution to the flux, the authors should show the premultiplied cospectrum and let the scale on the y-axis be linear. The authors have stated why they want to keep with the log-scale in Fig. 8, but this is not a strong argument. The figure is used in the context of explaining the underestimation of the momentum flux. If they want to impress with their signal quality, they should show the u-spectrum, which is a much stronger achievement, given that they measure at a near-vertical path. *In any way, they must still document that the "missing" 3% in uw flux stems from the range between 0.1 and 5Hz.* I don't believe that it does, but that rather the sum of the flux contribution in this range is much, much smaller. I believe that the difference come from the energy containing range, and if this is the case, then the H15 shows a substantial improvement of 3% in the uw comparison contrary to what the authors state now.

  "An analysis of the cospectra shows that the H15 correction somewhat distorts the expected −7/3 power-law behavior at very high frequencies in the inertial subrange (**Fehler! Verweisquelle konnte nicht gefunden werden.**). However, the values for f > 1 s⁻¹ are very small anyways and represent mostly white noise, which appears as horizontal line in spectral space. It can also be seen that the H15 correction slightly increases the cospectral energy across the entire range of frequencies. ==However, the too steep drop-off of the CSAT3B ensemble cospectrum is not improved effectively.=="

  Of course, H15 cannot be used for low-pass filtering corrections, it was never intended for this use and it is utterly strange that the authors use this argument for not using the H15. Low-pass filtering effects due to path-averaging should be compensated for with path-averaging corrections, please see next comment.

- Low-pass filtering correction/compensation for path-averaging:

  The Moore correction was not applied on the presented spectra but only on the standard deviations and u*. Therefore, it did not affect the calculation of the spectral ratio for this study. Moreover, the implementation of the Moore correction in the TK3 software is applied to all three velocity components and the effect on the resulting standard deviations is very small as a result of the large ratio between the measurement height and the path lengths of the two instruments (>250). Pena et al. also only applied the Horst and Oncley correction to the CSAT3 data for the lower measurement height (6.4 m) and not to ones with a larger measurement height (76 m)

  The authors apply an inaccurate low-pass filtering correction erroneously, and this is not acceptable in a study where effects as small as 1% are of importance. I asked the authors to change from the Moore correction, which is an inexact approximation for path-length averaging on the vertical velocity component, to the correction published by Horst and Oncley in 2006, which is a more exact correction determined explicitly for the CSAT sonic. The answer that this is of no importance and that the authors further have applied the approximate correction for all three velocity components is not reassuring, given that Horst and Oncley (2006) showed that the three velocity components are affected differently by the path-length correction.
  The authors should just do it right! Whereas the absolute size of the correction is likely negligible for large fluxes and higher wind speeds, it can be of importance for low-wind speed stable situations, and these situations were not included in the Pena et al 2019 paper. Hence, it is not obvious how large the correction is, based only on measurement height. In Pena et al (2019), the correction was applied for 6.5m and 16m but low-wind speed situations were not presented. They stated their results with and without path-averaging correction and it was clear what the effects were. This is quite different from the current study.
  The only alternative to applying the path-length correction by Horst and Oncley (2006) is to apply no correction for both statistics and spectra.

- I noted that the second reviewer asked the authors to refrain from being subjective in their judgement of the results, and found this a very good point. Yet, the authors still state in the abstract that the agreement is "very good". The view of "very good" can however be challenged. In example, for wind resource assessment in the field of wind energy, systematic errors of 1-2% in the mean wind speed are viewed as problematic and the CSAT sonic shows such wind direction dependent errors if one trusts the PTB lidar; around 4% for 160 deg. and -2% for 270 deg. (Fig 6). The authors should just state their results and remove the occurrences in the abstract of "very good".
  Further, unless the data are still affected by spikes, the result for sig_u must show the same directional dependency as that of the mean velocity. And if the sig_u shows a directional dependence, one wonders about the other two velocity components...

- lines 18-19, p. 15: Comment on citation to Hogstrom and Smedman (2004). If the authors read the whole Hogstrom and Smedman paper, it should be clear to them that the statement regarding transferability from wind tunnel to atmosphere is based on a speculation and not a proven result. Whereas wind tunnels for sure have their limitations, I believe that both Hogstrom and Smedman (2004) as well as the current authors are mistaken in their strong rejection of their usefulness. Whereas the difference in drag on cylinders in laminar and turbulent flows is indeed well-known, it should be stressed that wind tunnels are not entirely laminar, but flows in wind tunnel contain a large amount of very small scale turbulence. And it is the small-scale turbulence with similar or smaller length scale than the diameter of the cylinders

that matters for the drag on the cylinder. We will for sure not settle this issue here, but since the statement is strong and of very high importance for people working with wind tunnels, I would recommend to cite carefully and correctly. However, I agree with the authors that conclusions stated in the abstract of papers should be citable, so technically, the authors have a right to cite as they do.

---

## Author Response (AR2)

**Response to Report #2**

The original reviewer comments are in black font and our detailed responses use blue font.

I have read the author's comments to both reviewers and the revised manuscript and believe that this paper is acceptable for publication after the consideration of one minor comment:

In the revised manuscript on page 3, line 25-27, a new sentence has been added that states "This is important because horizontal wind velocities are usually much larger than vertical wind velocities, and a distorted measurement of the horizontal wind speed directly affects the vertical wind speed measurement". This statement is vague and could be misinterpreted.

In the context of this paragraph, this statement is probably meant to explain the advantages of the non-orthogonal CSAT3 versus previous anemometers that were orthogonal. Yet, for an orthogonal anemometer, this statement could be inaccurate because the horizontal and vertical measurements are entirely independent. It could be argued that flow distortions due to the horizontal transducers/structure might fundamentally distort the wind flow in all dimensions such that the vertical component is fundamentally changed, leading to an accurate measurement of a distorted vertical wind component. I'm not sure if this has ever been proven or disproven. Regardless, for an orthogonal anemometer, it is not clear that there is any correlation between horizontal and vertical measurement errors.

Regarding a non-orthogonal anemometer, it is important to note that all measurement errors originate in the transducer measurements. This means that while a distorted horizontal measurement will directly affect the transducer measurements, it might not affect the vertical wind speed measurement. This would occur if the measurement error only affects the horizontal components of the transducer measurements. In this case, while the transducers measurements would be affected, the vertical wind measurement (which is proportional to the sum of the three transducer measurements in a non-orthogonal anemometer) could still be unaffected.

I would ask the authors to consider revising this sentence.

Otherwise, well done.

John Frank

Thank you for this comment. It helped us to realize that this sentence can be misinterpreted, because it does not apply to orthogonal arrangements of sonic paths. Therefore, we have modified this sentence accordingly:

*"This is important because horizontal wind velocities are usually much larger than vertical wind velocities, and when using sonic anemometers with non-orthogonal paths, a distorted measurement of the horizontal wind speed directly affects the vertical wind speed measurement."*

**Response to Report #1**

Remaining criticism:

The authors have responded well to most of the criticism, and errors have been corrected, which is good. I remain truly impressed by the PTB lidar, but I still see a few potential errors in the paper.

• Concerning stating uncertainties in the PTB lidar. All papers should address the main uncertainties in the presented data; this is standard scientific good practice. Given that the authors present somewhat inconsistent results (see below), it would be interesting for the readers to learn, whether there is a possible weak point. In example, the authors write in the review answer that an error as small as 0.0034deg in the azimuth is a huge problem. What is their pointing accuracy and how does it translate to wind speed errors?

We believe this is a misinterpretation of our response. A misalignment of 0.0034° is not a huge problem for the PTB lidar. It is rather something that can be clearly detected by the PTB lidar.

• Abstract:

"Analysis of the corresponding cospectra showed that the CSAT3B underestimates this quantity systematically by about 3 % on average as a result of too steep a drop-off in the inertial subrange."
The authors still claim that the difference in momentum flux between the PTB lidar and the sonic anemometer is due to the clearly seen difference in the spectra between 0.1 and 5Hz in Figure 8. However, as stated in my original review, because of the double log-axis plot, this way of illustrating error could be very misleading. In order to let the reader be able to see the frequency dependent contribution to the flux, the authors should show the premultiplied cospectrum and let the scale on the y-axis be linear. The authors have stated why they want to keep with the log-scale in Fig. 8, but this is not a strong argument. The figure is used in the context of explaining the underestimation of the momentum flux. If they want to impress with their signal quality, they should show the u-spectrum, which is a much stronger achievement, given that they measure at a near-vertical path. *In any way, they must still document that the "missing" 3% in uw flux stems from the range between 0.1 and 5Hz.* I don't believe that it does, but that rather the sum of the flux contribution in this range is much, much smaller. I believe that the difference come from the energy containing range, and if this is the case, then the H15 shows a substantial improvement of 3% in the uw comparison contrary to what the authors state now.

Of course, H15 cannot be used for low-pass filtering corrections, it was never intended for this use and it is utterly strange that the authors use this argument for not using the H15. Low-pass filtering effects due to path-averaging should be compensated for with path-averaging corrections, please see next comment.

It is correct that, in contrast to a double-log plot, a semi-log plot of the pre-multiplied cospectra has the property that the area under any portion of the curve is proportional to the covariance. Such a representation of the same data is presented below:

[Figure]

In this figure, you can also see the same behavior that we describe in the manuscript, which can be summarized in two main points:

1. The underestimation of the u'w' (and hence u*) by the CSAT3B compared to the PTB lidar stems mostly from the range of frequencies between 0.1 Hz and 5 Hz. At lower frequencies, there seem to be no systematic differences between the between the CSAT3B and the PTB lidar in the uw cospectra. For illustration of this statement, please see also the difference plot below:

[Figure]

2. The H15 correction leads to an increase in cospectral energy density across all frequencies, but it does not effectively correct for this cospectral loss at high frequencies, which is the main reason for the discrepancies between the CSAT3B and the PTB lidar.

Although the semi-log plot has the above-mentioned advantages, we would still like to retain the double-log version of the plot in the manuscript, because it has the advantage that the power law appears in a straight line, and this figure illustrates that the PTB lidar follow the theoretical power law much better than the CSAT3B. This is important, because it supports our choice that the PTB lidar can be viewed as reference instrument in this comparison. In order to make this point clearer and more precise we reformulated this sentence in the abstract as follows:

*"Analysis of the corresponding cospectra showed that the CSAT3B underestimates this quantity systematically by about 3 % on average as a result of cospectral losses in the frequency range between 0.1 $s^{-1}$ and 5 $s^{-1}$."*

We have also modified the paragraph describing the cospectra in the manuscript for further clarification.

• Low-pass filtering correction/compensation for path-averaging:
The authors apply an inaccurate low-pass filtering correction erroneously, and this is not acceptable in a study where effects as small as 1% are of importance. I asked the authors to change from the Moore correction, which is an inexact approximation for path-length averaging on the vertical velocity component, to the correction published by Horst and Oncley in 2006, which is a more exact correction determined explicitly for the CSAT sonic. The answer that this is of no importance and that the authors further have applied the approximate correction for all three velocity components is not reassuring, given that Horst and Oncley (2006) showed that the three velocity components are affected differently by the path-length correction.
The authors should just do it right! Whereas the absolute size of the correction is likely negligible for large fluxes and higher wind speeds, it can be of importance for low-wind speed stable situations, and these situations were not included in the Pena et al 2019 paper. Hence, it is not obvious how large the correction is, based only on measurement height. In Pena et al (2019), the correction was applied for 6.5m and 16m but low-wind speed situations were not presented. They stated their results with and without path-averaging correction and it was clear what the effects were. This is quite different from the current study.
The only alternative to applying the path-length correction by Horst and Oncley (2006) is to apply no correction for both statistics and spectra.

I assume we all agree that any correction is just a model of the reality, and the difference between different correction models is therefore just gradual and not categorical, depending on the level of simplification and generalization. Hence, we would not agree with a statement that says that one correction is absolutely correct and the other one is inaccurate. The model of Horst and Oncley (2006) also has some simplifications and generalizations, admittedly less than the Moore correction but it is still just a model. In any case, we aim to avoid confusion and we want to present our results in a way that is as transparent as possible. Therefore, we follow the advice of this reviewer and decided to add a discussion of the results without low-pass filtering correction for path-length averaging for the turbulence statistics and also present the results of the statistical comparison in Table 1. None of the conclusions are affected because the differences due to the low-pass filtering correction are much smaller than the differences caused by the H15 correction.

• I noted that the second reviewer asked the authors to refrain from being subjective in their judgement of the results, and found this a very good point. Yet, the authors still state in the abstract that the agreement is "very good". The view of "very good" can however be challenged. In example, for wind resource assessment in the field of wind energy, systematic errors of 1-2% in the mean wind speed are viewed as problematic and the CSAT sonic shows such wind direction dependent errors if one trusts the PTB lidar; around 4% for 160 deg. and -2% for 270 deg. (Fig 6). The authors should just state their results and remove the occurrences in the abstract of "very good".
We have removed qualitative statements, such as "very good", from the abstract and now just report the results in quantitative terms in the abstract.

Further, unless the data are still affected by spikes, the result for sig_u must show the same directional dependency as that of the mean velocity. And if the sig_u shows a directional dependence, one wonders about the other two velocity components...

These larger difference in u_bar can already be seen in the scatter plot (Fig 5) and that is the reason why we investigated their wind-direction dependence. The scatter plots of sig_u, sig_v and sig_w are provided in the manuscript (or in the appendix), and they do not show such larger differences for a number of data points, which would warrant further investigation.

• lines 18-19, p. 15: Comment on citation to Hogstrom and Smedman (2004). If the authors read the whole Hogstrom and Smedman paper, it should be clear to them that the statement regarding transferability from wind tunnel to atmosphere is based on a speculation and not a proven result. Whereas wind tunnels for sure have their limitations, I believe that both Hogstrom and Smedman (2004) as well as the current authors are mistaken in their strong rejection of their usefulness. Whereas the difference in drag on cylinders in laminar and turbulent flows is indeed well-known, it should be stressed that wind tunnels are not entirely laminar, but flows in wind tunnel contain a large amount of very small scale turbulence. And it is the small-scale turbulence with similar or smaller length scale than the diameter of the cylinders that matters for the drag on the cylinder. We will for sure not settle this issue here, but since the statement is strong and of very high importance for people working with wind tunnels, I would recommend to cite carefully and correctly. However, I agree with the authors that conclusions stated in the abstract of papers should be citable, so technically, the authors have a right to cite as they do.

It is clear that the results from this field intercomparison between a sonic anemometer and a 3D Doppler lidar partially disagree with the results of from wind tunnels. The field intercomparison indicates much smaller flow distortion errors. The question is which type of experiments has more validity in principle. Since this lidar experiment was conducted under typical field conditions, where sonic anemometers are normally deployed, we are confident that these results obtained in the field are more reliable. This does not mean that wind-tunnel experiments are not useful in general, and we do not write this in the manuscript. For example, we have also conducted a wind-tunnel experiment in a previous study to assess the performance of the PTB lidar against an LDA, which is no problem at all because the lidar is free of flow distortion anyways and the comparison can therefore be assumed to be independent of the Reynolds number. We also agree that there are ways to introduce turbulence in wind tunnels as well. Nevertheless, the Reynolds numbers are still smaller by several orders of magnitude. Therefore, we believe that this is the best explanation of the observed discrepancies between wind-tunnel and field experiments. For clarification, this senctence has been added to the manuscript:

*"Therefore, we believe that this explains the differences between our field study and previous wind-tunnel based and LES-based experiments, and we expect that the field experiment has more validity in principle, since sonic anemometers are normally used in the field."*

What we actually also want to say is that it is generally better to avoid a flow-distortion correction through a clever design of the sensor than relying on a wind-tunnel based correction, which might not be directly transferable to the real world. For clarification, we added a sentence along these lines to the manuscript:

*"Moreover, it is generally preferable to minimize flow-distortion errors to begin with through an optimized design of the instrument, e.g. by increasing the ratio between path length and transducer*

*diameter, than relying on the transferability of wind-tunnel based correction models to real-world conditions."*

Please note that our statements in this manuscript on wind tunnel corrections are not really that strong. We use rather weak formulations such as "might not be applicable in the turbulent free atmosphere" or "it is problematic to transfer quasi-laminar wind-tunnel calibrations". Please also note that we don't write that wind tunnels are purely laminar but "quasi-laminar".